# Tex4D: Zero-shot 4D Scene Texturing with Video Diffusion Models

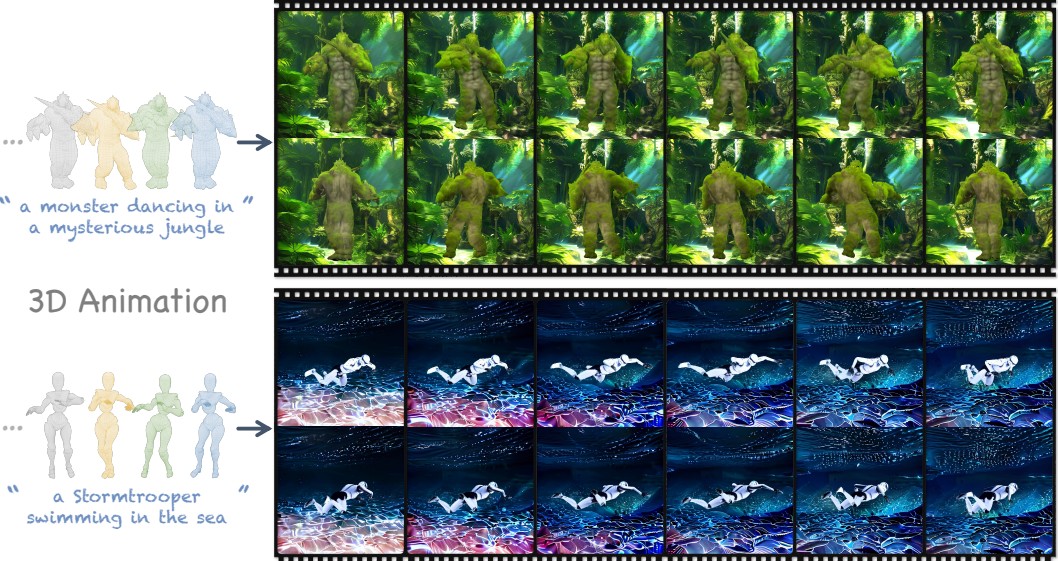

Figure 1. Given an untextured mesh sequence and a text prompt as inputs (Left), **Tex4D** generates multi-view, temporally consistent textures along with a dynamic background. On the right, we show renderings of the textured meshes from two different perspectives. Zoom in to view the texture details.

## Abstract

3D meshes are widely used in computer vision and graphics because of their efficiency in animation and minimal memory footprint. They are extensively employed in movies, games, AR, and VR, leading to the creation of a vast number of mesh sequences. However, creating temporally consistent and realistic textures for these mesh sequences to model the appearance transformations remains labor-intensive for professional artists. On the other hand, video diffusion models have demonstrated remarkable capabilities in text-driven video generation, enabling users to create countless video clips based solely on their imagination. Despite their strengths, these models often lack 3D geometry awareness for fine-grained video control and struggle with achieving multi-view consistent texturing for 3D mesh sequences. In this work, we present **Tex4D**, a zero-shot approach that integrates inherent 3D geometry knowledge from mesh sequences with the expressiveness of video diffusion models. Given an untextured mesh sequence and a text prompt as inputs, our method enhances multi-view consistency by synchronizing the diffusion process across different views through latent aggregation in the UV space. To ensure temporal consistency such as lighting changes, wrinkles, and appearance transformations, we leverage prior knowledge from a conditional video generation model for texture synthesis. However, straightforwardly combining the video diffusion model and the UV texture aggregation leads to blurry results. We analyze the underlying causes and propose a simple yet effective modification to the DDIM sampling process to address this issue. Additionally, we introduce a reference latent texture to strengthen the correlation between frames during the denoising process. To the best of our knowledge, Tex4D is the first method specifically designed for 4D scene texturing. Extensive experiments demonstrate its superiority in producing multi-view and multi-frame consistent videos based on untextured mesh sequences.

# 1 INTRODUCTION

3D meshes are widely used in modeling, computer-aided design (CAD), animation, and computer graphics due to their low memory footprint and efficiency in animation. Visual artists, game designers, and movie creators build numerous animated mesh sequences for visual applications. However, creating vivid videos involves complex post-processing steps, such as lighting controls and appearance transformations. These steps are labor-intensive and require specialized expertise by artists.

On the other hand, recent advancements in generative models have democratized content creation and demonstrated impressive performance in image and video synthesis. For instance, video generation models (Ho et al., 2022; Esser et al., 2023; Li et al., 2023; He et al., 2022; Yu et al., 2023a; Zhou et al., 2022; Hong et al., 2022; Yang et al., 2024; Zhang et al., 2023b; Xing et al., 2023; Chen et al., 2023c; 2024) trained on large-scale video datasets (Bain et al., 2021; Schuhmann et al., 2021) allow users to create realistic video clips from various inputs such as text prompts, images, or geometric conditions. However, these text-to-video generation models, which are trained solely on 2D data, often struggle with spatial consistency when applied to multi-view image generation (Tang et al., 2023; Shi et al., 2023b; Liu et al., 2023a; Weng et al., 2023; Long et al., 2023; Shi et al., 2023a; Kwak et al., 2023; Tang et al., 2024; Voleti et al., 2024) or 3D object texturing (Cao et al., 2023; Liu et al., 2023b; Richardson et al., 2023; Chen et al., 2023b; Huo et al., 2024).

To address these limitations, two main approaches have been developed. One approach (Richardson et al., 2023; Chen et al., 2023b; Cao et al., 2023) focuses on resolving multi-view inconsistency in static 3D object texturing by synchronizing multi-view image diffusion processes and enforcing UV space consistency. While these methods produce multi-view consistent textures for static 3D objects, they do not address the challenge of generating temporally consistent textures for mesh sequences. Another approach (Guo et al., 2023a; Lin et al., 2024; Peng et al., 2024) aims to generate temporally consistent video clips based on the rendering (e.g., depth, normal or UV maps) of an untextured mesh sequence. To encourage temporal consistency, these methods modify the attention mechanism in 2D diffusion models and utilize inherent correspondences in a mesh sequence to facilitate feature synchronization between frames. Although these techniques can be adapted for multi-view image generation by treating camera pose movement as temporal motion, they usually produce inconsistent 3D texturing due to insufficient exploitation of 3D geometry priors.

In this paper, we introduce a novel task: 4D scene texturing. Given an animated untextured 3D mesh sequence and a text prompt, our goal is to generate textures that are both temporally and multi-view consistent. Our objective is to texture 4D scenes while capturing temporal variations, such as lighting changes and wrinkles, to produce vivid visual results—a key requirement in downstream tasks like character generation. Different from existing works, we fully leverage 3D geometry knowledge from the mesh sequence to enforce multi-view consistency. Specifically, we develop a method that synchronizes the diffusion process from different views through latent aggregation in the UV space. To ensure temporal consistency, we employ prior knowledge from a conditional video generation model for texture synthesis and introduce a reference latent texture to enhance frame-to-frame correlations during the denoising process. However, naively integrating the UV texture aggregation into the video diffusion process causes the variance shift problem, leading to blurry results. To resolve this issue, we propose a simple yet effective modification to the DDIM (Song et al., 2020) sampling process by uniformly transforming the equation. Additionally, we propose to synthesize a dynamic background along with the textures of the given mesh sequences, which not only creates a complete 4D scene but also fully exploits the prior knowledge embedded in the video diffusion model. Our method is computationally efficient thanks to its zero-shot nature. The textured mesh sequence can be rendered from any camera view, thus supporting a wide range of applications in content creation.

We evaluate our method on various mesh sequences with key contributions as follows:

- We present **Tex4D**, a zero-shot pipeline for generating high-fidelity textures that are temporally and multi-view consistent, utilizing text-to-video diffusion models and mesh sequence controls.
- We develop a simple and effective modification to the DDIM sampling process to address the variance shift issue caused by multi-view texture aggregation.
- We introduce a reference UV blending mechanism to establish correlations during the denoising steps, addressing self-occlusions, and synchronizing the diffusion process in invisible regions.
- Our method is not only computationally efficient, but also demonstrates comparable if not superior performance to various state-of-the-art baselines.

## 2 RELATED WORK

**Video Stylization and Editing**   Text-to-video diffusion models have shown remarkable performance in the field of video generation. These models learn motions and dynamics from large-scale video datasets using 3D-UNet to create high-quality, realistic, and temporally coherent videos. Although these approaches show compelling results, the generated videos lack fine-grained control, inhibiting their application in stylization and editing. To solve this issue, inspired by ControlNet (Zhang et al., 2023a), SparseCtrl (Guo et al., 2023a) trains a sparse encoder from scratch using frame masks and sparse conditioning images as input to guide a pre-trained video diffusion model. CTRL-Adapter (Lin et al., 2024) proposes a trainable intermediate adapter to efficiently connect the features between ControlNet and video diffusion models.

Meanwhile, Tumanyan et al. (2023) observed that the spatial features of T2I models play an influential role in determining the structure and appearance, Text2Video-Zero (Khachatryan et al., 2023) uses a frame-warping method to animate the foreground object by T2I models and Wu et al. (2023); Ceylan et al. (2023); Qi et al. (2023) propose utilizing self-attention injection and cross-frame attention to generate stylized and temporally consistent video using DDIM inversion (Song et al., 2020). Subsequently, numerous works (Zhang et al., 2023c; Cai et al., 2024; Yang et al., 2023; Geyer et al., 2023; Eldesokey & Wonka, 2024) generate temporally consistent videos utilizing T2I diffusion models by spatial latent alignment without training. However, the synthesized videos usually show flickerings due to the empirical correspondences, such as feature embedding distances and UV maps, which are insufficient to express the continuous content in the latent space. Another line of work (Singer et al., 2022; Bar-Tal et al., 2022; Blattmann et al., 2023; Xu et al., 2024; Guo et al., 2023b) is to train additional modules on large-scale video datasets to construct feature mappings, for example, Text2LIVE (Bar-Tal et al., 2022) applies test-time training with the CLIP loss, and MagicAnimate (Xu et al., 2024) introduced an appearance encoder to retain intricate clothes details.

**Texture Synthesis**   With the rapid development of foundation models, researchers have focused on applying their generation capability and adaptability to simplify the process of designing textures and reduce the expertise required. To incorporate the result 3D content with prior knowledge, earlier works (Khalid et al., 2022; Michel et al., 2021; Chen et al., 2022) jointly optimize the meshes and textures from scratch with the simple semantic loss from the pre-trained CLIP (Radford et al., 2021) to encourage the 3D alignment between the generated results and the semantic priors. However, the results show apparent artifacts and distortion because the semantic feature cannot provide fine-grained supervision during the generation of 3D content.

DreamFusion (Poole et al., 2022) and similar models (Lin et al., 2023; Wang et al., 2023; Po & Wetzstein, 2024; Metzer et al., 2022; Chen et al., 2023a) distill the learned 2D diffusion priors from the pre-trained diffusion models (Rombach et al., 2021) to synthesize the 3D content by Score Distillation Sampling (SDS). These methods render 2D projections of the 3D asset parameters and compare them against reference images, iteratively refining the 3D asset parameters to minimize the discrepancy of the target distribution of 3D shapes learned by the diffusion model. Although these approaches enable people without expertise to generate detailed 3D content by textual prompt, their results are typically over-saturated and over-smoothed, hindering their application in actual cases. Another line of optimization-based methods (Yu et al., 2023b; Zeng et al., 2024; Bensadoun et al., 2024) turned to fuse 3D shape information, such as vertex positions, depth maps and normal maps, with the pre-trained diffusion model by training separate modules on 3D datasets. Still, they require a specific UV layout process to achieve plausible results.

Recently, TexFusion (Cao et al., 2023) and numerous zero-shot methods (Liu et al., 2023b; Richardson et al., 2023; Chen et al., 2023b; Huo et al., 2024) have shown significant success in generating globally consistent textures without additional 3D datasets. Based on depth-aware diffusion models, they sequentially inpaint the latents in the UV domain to ensure the spatial consistency of latents observed across different views. Then, they decode the latents from multiple views and finally synthesize the RGB texture through backprojection.

However, these methods primarily focus on generating static 3D assets and do not account for temporal changes in the final visual presentation, such as in videos. Our work introduces a zero-shot framework that enables multi-view consistent video generation based on animated meshes, which is effective in capturing temporal variations. To the best of our knowledge, this is the first approach to synthesize multi-view and multi-frame consistent textures for mesh sequences.

## 3 PRELIMINARIES

**Video Diffusion Prior.** In this paper, we adopt CTRL-Adapter (Lin et al., 2024) as our prior model to provide dynamic information. CTRL-Adapter aims to adapt a pre-trained text-to-video diffusion model to condition various types of images such as depth or normal map sequences. The key idea behind CTRL-Adapter is to leverage a pre-trained ControlNet (Zhang et al., 2023a) and to align its latents with those of the video diffusion model through a learnable mapping module. Intuitively, the video diffusion model generates temporally consistent video frames that capture dynamic elements like character motions and environmental lighting, while the ControlNet further enhances this capability by allowing the model to condition on geometric information, such as depth and normal map sequences. This makes CTRL-Adapter particularly effective in providing a temporally consistent texture prior for our 4D scene texturing task. Specifically, we leverage the depth-conditioned CTRL-Adapter model. Given a sequence of depth images denoted as $\{D_1, ..., D_K\}$ and a text prompt $\mathcal{P}$, CTRL-Adapter (denoted as $\mathcal{C}$) synthesizes a frame sequence $F$ by $F = \mathcal{C}(\{D_1, ..., D_K\}, \mathcal{P})$.

**DDIM Sampling.** DDIM (Song et al., 2020) is a widely used sampling method in diffusion models due to its superior efficiency and deterministic nature compared to DDPM (Ho et al., 2020). To enhance numerical stability and prevent temporal color shifts in Video Diffusion Models (VDMs), numerous models (Zhang et al., 2023b; Ho et al., 2022) employ a learning-based sampling technique known as v-prediction (Salimans & Ho, 2022). At each denoising step, the DDIM sampling process for the latents (denoted as $z_t$) can be described as follows:

$$z_{t-1} = \sqrt{\alpha_{t-1}} \cdot \hat{z}_0(z_t) + \sqrt{1 - \alpha_{t-1}} \cdot \epsilon_\theta(z_t), \tag{1}$$

$$\hat{z}_0(z_t) = \frac{z_t - \sqrt{1 - \alpha_t} \cdot \epsilon_\theta}{\sqrt{\alpha_t}}, \quad \epsilon_\theta(z_t) = \epsilon_\theta, \tag{2}$$

where $\alpha_t$ represents the noise variance at time step $t$, $\epsilon_\theta$ is the estimated noise from the U-Net denoising module, which is expected to follow $\mathcal{N}(0, \mathcal{I})$, and $\hat{z}_0(z_t)$ denotes the predicted original sample (i.e., the latents at timestep 0). After the v-parameterization, the predicted original sample $\hat{z}_0(z_t)$ and the predicted epsilon $\epsilon_\theta(z_t)$ are computed as follows:

$$\hat{z}_0(z_t) = \sqrt{\alpha_t} \cdot z_t - \sqrt{1 - \alpha_t} \cdot \epsilon_\theta, \quad \epsilon_\theta(z_t) = \sqrt{\alpha_t} \cdot \epsilon_\theta + \sqrt{1 - \alpha_t} \cdot z_t. \tag{3}$$

In this paper, we leverage an enhanced DDIM sampling process in video diffusion models, along with a multi-view consistent texture aggregation mechanism to synthesize 4D textures.

## 4 METHOD

Given an untextured mesh animation and a text prompt, our goal is to generate multi-view and multi-frame consistent texture for each mesh that aligns with both the text description and motion cues, meanwhile capturing the dynamics from video diffusion models. To optimize computational efficiency, instead of processing all video frames, we uniformly sample $K$ key frames from the video and synthesize textures specifically for these key frames. The textures for the remaining frames are then generated by interpolating the key frame textures. Formally, given $K$ animated meshes at the key frames ($\{M_1, ..., M_K\}$), along with a text description $\mathcal{P}$, our method produces a sequence of temporally and spatially consistent UV maps denoted as $\{UV_1, ..., UV_K\}$, in a zero-shot manner.

Previous texture generation methods (Richardson et al., 2023; Chen et al., 2023b; Cao et al., 2023) typically inpaint and update textures sequentially using pre-defined camera views in an incremental manner. However, these approaches rely on view-dependent depth conditions and lack global spatial consistency, often resulting in visible discontinuities in the assembled texture map. This issue arises from error accumulation during the autoregressive view update process, as noted by Bensadoun et al. (2024). To resolve these issues, rather than processing each view independently, recent methods (Liu et al., 2023b; Huo et al., 2024; Zhang et al., 2024) propose to generate multi-view textures simultaneously through diffusion, and then aggregate them in the UV space at each diffusion step. In this work, we similarly leverage the UV space as an intermediate representation to ensure multi-view consistency during texture generation.

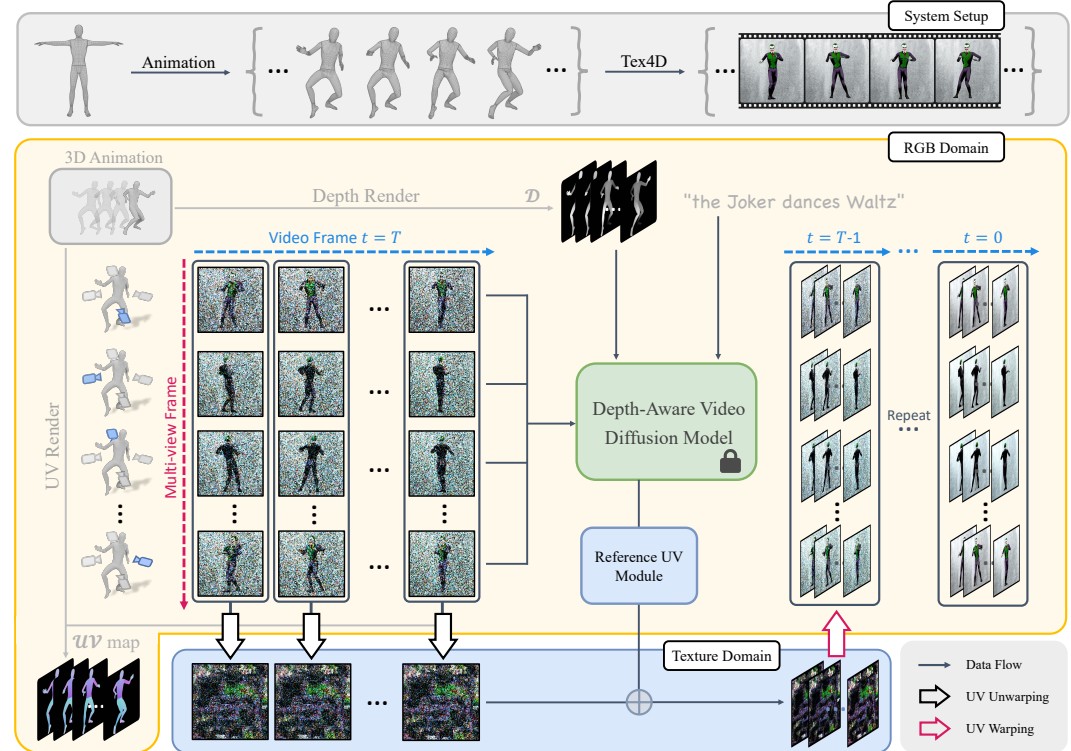

Figure 2. **Overview of our pipeline.** Given a mesh sequence and a text prompt as inputs, Tex4D generates a UV-parameterized texture sequence that is both globally and temporally consistent, aligning with the prompt and the mesh sequence. We sample multi-view video sequences using a depth-aware video diffusion model. At each diffusion step, latent views are aggregated into UV space, followed by multi-view latent texture diffusion to ensure global consistency. To maintain temporal coherence and address self-occlusions, a Reference UV Blending module is applied at the end of each step. Finally, the latent textures are back-projected and decoded to produce RGB textures for each frame.

## 4.1 OVERVIEW

As shown in Fig. 2, given a sequence of $K$ meshes, we start by rendering the mesh at $V$ predefined, uniformly sampled camera poses to obtain multi-view depth images (denoted as $\{D_{1,1}, ..., D_{1,K}, D_{2,1}..., D_{V,K}\}$), which serve as the geometric conditions. To generate textures for each mesh, we initialize $V \times K$ noise images sampled from a Normal distribution (denoted as $\{z^{1,1}, ..., z^{1,K}, z^{2,1}, ..., z^{V,K}\}$). Additionally, we initialize an extra noise map sequence $\{z_b^1, ..., z_b^K\}$ for the backgrounds learning. This noise map corresponds to the texture of a plane mesh that is composited with the foreground object at each diffusion step (See Sec. 4.3). Next, for each view $v \in \{1, ..., V\}$, we apply the video diffusion model (Lin et al., 2024) discussed in Sec. 3 to simultaneously denoise all latents and obtain multi-frame consistent images as $\{I^{1,v}, ..., I^{K,v}\} = \mathcal{C}(\{D_{1,v}, ..., D_{K,v}\}, \mathcal{P})$, where $\mathcal{P}$ is the provided text prompt. Finally, we un-project and aggregate all denoised multi-view images for each mesh to formulate temporally consistent UV textures.

However, applying the video diffusion model independently to each camera view often results in multi-view inconsistencies. Inspired by (Liu et al., 2023b; Huo et al., 2024; Zhang et al., 2024), we aggregate the multi-view latents of each mesh in the UV space to merge observations across different views at each denoising step. We then render latent from the latent texture to ensure multi-view consistency. To simultaneously generate a dynamic background and fully exploit prior in the video diffusion model, we composite the rendered foreground latents with the background latents at each diffusion step. This aggregation process is discussed in detail in Sec. 4.2. Nonetheless, such a simple aggregation method introduces blurriness in the final results. In Sec. 4.3, we analyze the underlying causes and propose a simple yet effective method to enhance the denoising process. Additionally, we create and leverage a reference UV to handle self-occlusions and further improve temporal consistency in Sec. 4.4.

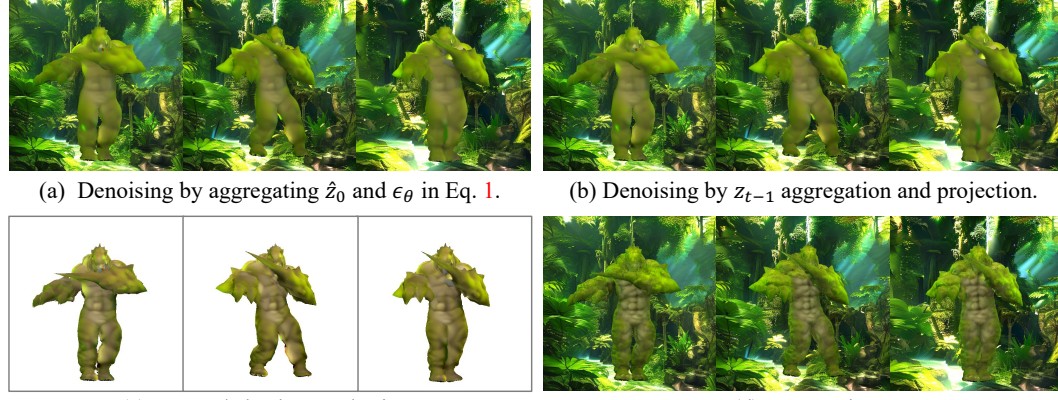

(a) Denoising by aggregating $\hat{z}_0$ and $\epsilon_\theta$ in Eq. 1.     (b) Denoising by $z_{t-1}$ aggregation and projection.

(c) Ours w/o background priors.     (d) Our results.

Figure 3. **Ablation studies on the multi-view denoising algorithm and backgrounds.** (a) Aggregating $\hat{z}_0(z_t)$, $\epsilon_\theta(z_t)$ in Eq. 1 into UV space. (b) Aggregating $z_{t-1}$ in Eq. 1 into UV space. (c) Replacing learnable background with white background. (d) Our results. See Sec. 5.3 for details.

## 4.2 MULTI-VIEW LATENTS AGGREGATION IN THE UV SPACE

We describe how to aggregate multi-view latent maps in the UV space. Taking frame $k \in \{1, ..., K\}$ as an example, we aggregate the multi-view latents $\{z^{1,k}, ..., z^{V,k}\}$ in the UV space by:

$$\mathcal{T}^k\left(z^k\right) = \frac{\sum_{v=1}^{V} \mathcal{R}^{-1}(z^{v,k}, c_v) \odot \cos\left(\theta^v\right)^\alpha}{\sum_{v=1}^{V} \cos\left(\theta^v\right)^\alpha}, \tag{4}$$

where $\mathcal{R}^{-1}$ represents the inverse rendering operator that un-projects the latents to the UV space, thus $\mathcal{R}^{-1}(z^{v,k}, c_v)$ produces a partial latent UV texture from view $v$, $\cos(\theta^v)$ is the cosine map buffered by the geometry shader, recording the cosine value between the view direction and the surface normal for each pixel, $\alpha$ is a scaling factor, and $c_v$ denotes one of the predefined cameras. After multi-view latents aggregation, we obtain multi-view consistent latents by rendering the aggregated UV latent map using $\tilde{z}^{v,k} = \mathcal{R}\left(\mathcal{T}^k; c_v\right)$, where $\mathcal{R}$ is the rendering operation.

## 4.3 MULTI-FRAME CONSISTENT TEXTURE GENERATION

The aggregation process discussed above yields multi-view consistent latents $\{\tilde{z}^{v,k}\}$ for the subsequent denoising steps. However, this simple aggregation and projection strategy leads to a blurry appearance as shown in Fig. 3(b). This issue arises primarily because the aggregation process depicted in Eq. 4 derails the DDIM denoise process. Specifically, the estimated noise $\epsilon_\theta(z_t)$ for each step in Eq. 1 is expected to follow $\mathcal{N}(0, \mathcal{I})$, but Eq. 4 indicates that after aggregating multi-view latents, the expected norm of variance of the noise distribution would be less than $\mathcal{I}$. We denote this as the "variance shift" issue caused by the texture aggregation.

To resolve this issue, we rewrite the estimated noise $\epsilon_\theta$ for a latent as the combination of the t-step latent $z_t$ and the estimated latent $\hat{z}_0(z_t)$ at step 0, thus the v-parameterized predicted epsilon $\epsilon_\theta(z_t)$ in Eq. 3 can be equally expressed as follow:

$$\begin{aligned} \epsilon_\theta &= \left(\sqrt{\alpha_t} \cdot z_t - \hat{z}_0(z_t)\right) / \sqrt{1 - \alpha_t} \\ \epsilon_\theta(z_t) &= \sqrt{\alpha_t} \cdot \epsilon_\theta + \sqrt{1 - \alpha_t} \cdot z_t \\ &= \sqrt{\frac{\alpha_t}{1 - \alpha_t}} \cdot \left(\sqrt{\alpha_t} z_t - \hat{z}_0(z_t)\right) + \sqrt{1 - \alpha_t} \cdot z_t. \end{aligned} \tag{5}$$

In practice, we carry out this denoising technique in the UV space. Specifically, we first compute the original texture map (i.e., texture map at step 0, denoted as $\hat{\mathcal{T}}_0$) by aggregating the predicted original multi-view image latents through Eq. 4. The noisy latent texture map at time step $t$ (denoted as $\mathcal{T}_t$) can be similarly computed. We then run one desnoising step by:

$$\mathcal{T}_{t-1} = \sqrt{\alpha_{t-1}} \cdot \hat{\mathcal{T}}_0 + \sqrt{1 - \alpha_{t-1}} \left( \sqrt{\frac{\alpha_t}{1 - \alpha_t}} \cdot (\sqrt{\alpha_t} \mathcal{T}_t - \hat{\mathcal{T}}_0) + \sqrt{1 - \alpha_t} \cdot \mathcal{T}_t \right). \tag{6}$$

Through experimentation, we observe that background optimization plays a crucial role in fully exploiting the prior within the video diffusion model. As shown in Fig. 3(c), using a simple white background leads to blurry results. This may be attributed to a mismatch between the white-background images and the training dataset, which likely contains fewer such examples, affecting the denoising process. To resolve this issue, we compute the final latents as the combination of the foreground latent $\tilde{z}_{t-1}$ projected from the aggregated UV latents and the residual background latent $z_{b,t-1}$ denoised by diffusion models. Specifically, we composite the estimated latents in the $t-1$ step as follows:

$$z_{t-1} = \tilde{z}_{t-1} \odot \mathcal{M}_{\text{fg}} + z_{b,t-1} \odot (1 - \mathcal{M}_{\text{fg}}), \quad \tilde{z}_{t-1}, \mathcal{M}_{\text{fg}} = \mathcal{R}(\mathcal{T}_{t-1}; c_v), \tag{7}$$

where $\mathcal{M}_{\text{fg}}$ represents the foreground mask of the mesh.

To summarize, our diffusion process starts with $K \times (V+1)$ randomly initialized noise maps sampled (i.e., $\{z_T^{1,k}, \ldots, z_T^{V,k}\}$, for foreground, $\{z_b^1, \ldots, z_b^K\}$ for background) and denoise them into images simultaneously. At each denoising step $t$ with the key frame $k$, we derive the estimated noises $\{\epsilon_{t-1}^{1,k}, \ldots, \epsilon_{t-1}^{V,k}\}$ using the video diffusion model and calculate the estimated original latent $\{\hat{z}_0^{1,k}, \ldots, \hat{z}_0^{V,k}\}$ by Eq. 2. Then, we use Eq. 4 to aggregate the latents onto UV space. Next, we utilize Eq. 6 to take the diffusion step in the UV space, and render the synchronized latents $\{\tilde{z}_{t-1}^{1,k}, \ldots, \tilde{z}_{t-1}^{V,k}\}$ from latent UVs $\{\mathcal{T}_{t-1}^1, \ldots, \mathcal{T}_{t-1}^K\}$ to ensure multi-view consistency. Finally, we composite the denoised latent with the latents at step $t-1$ according to foreground masks by Eq. 7.

## 4.4 REFERENCE UV BLENDING

While the video diffusion model ensures temporal consistency for latents from each view, consistency can sometimes diminish after aggregation in the texture domain. This issue primarily stems from the view-dependent nature of the depth conditions and the limited resolution of latents, which can lead to distortions when features from different camera angles are combined onto the UV texture. Additionally, self-occlusion during mesh animation often results in a loss of information in invisible regions.

To address these challenges, we introduce a reference UV map to provide additional correlations between latent textures across frames. Specifically, the reference UV map is constructed by sequentially combining latent textures over time, with each new texture filling only the empty texels of the reference UV map. Then, each texture is blended using the reference UV $\mathcal{T}_{\mathcal{UV}}$ with a mask $\mathcal{M}_{\mathcal{UV}}$ that labels the visible region:

$$\mathcal{T}_t^k = \left((1 - \lambda) \cdot \mathcal{T}_t^k + \lambda \cdot \mathcal{T}_{\mathcal{UV}}\right) \odot \mathcal{M}_{\mathcal{UV}}^k + \mathcal{T}_{\mathcal{UV}} \odot \left(1 - \mathcal{M}_{\mathcal{UV}}^k\right), \tag{8}$$

where $\lambda$ is the blending weight for the reference UV in the visible region, while the invisible region is simply replaced with the reference texture. We empirically set the blending weight to $0.2$ during our experiments.

## 5 EXPERIMENTS

**Datasets.** We sourced our datasets from two primary repositories: human motion diffusion outputs and the Mixamo[1] and Sketchfab[2] websites. We employed the text-to-motion diffusion model (HDM) (Tevet et al., 2023) to compare our approach with LatentMan (Eldesokey & Wonka, 2024), as LatentMan requires the SMPL model (Loper et al., 2015) to get corresponding features. For comparison with Generative Rendering (Cai et al., 2024), we obtained animated characters from the Mixamo platform and rendered them with different motions. Specifically, we first used Blender Community (2024) to extract meshes, joints, skinning weights, and animation data from the FBX files. Then, we applied linear blend skinning to animate the meshes. For meshes without UV maps, we utilized XATLAS to parameterize the mesh and unwrap the UVs.

**Baselines.** To the best of our knowledge, no existing studies directly address the task of multi-view consistent video generation guided by untextured mesh sequences, as our method does. Consequently, we adapted six recent methods and rendered the input (untextured mesh renders and

---

[1]https://www.mixamo.com/
[2]https://sketchfab.com/

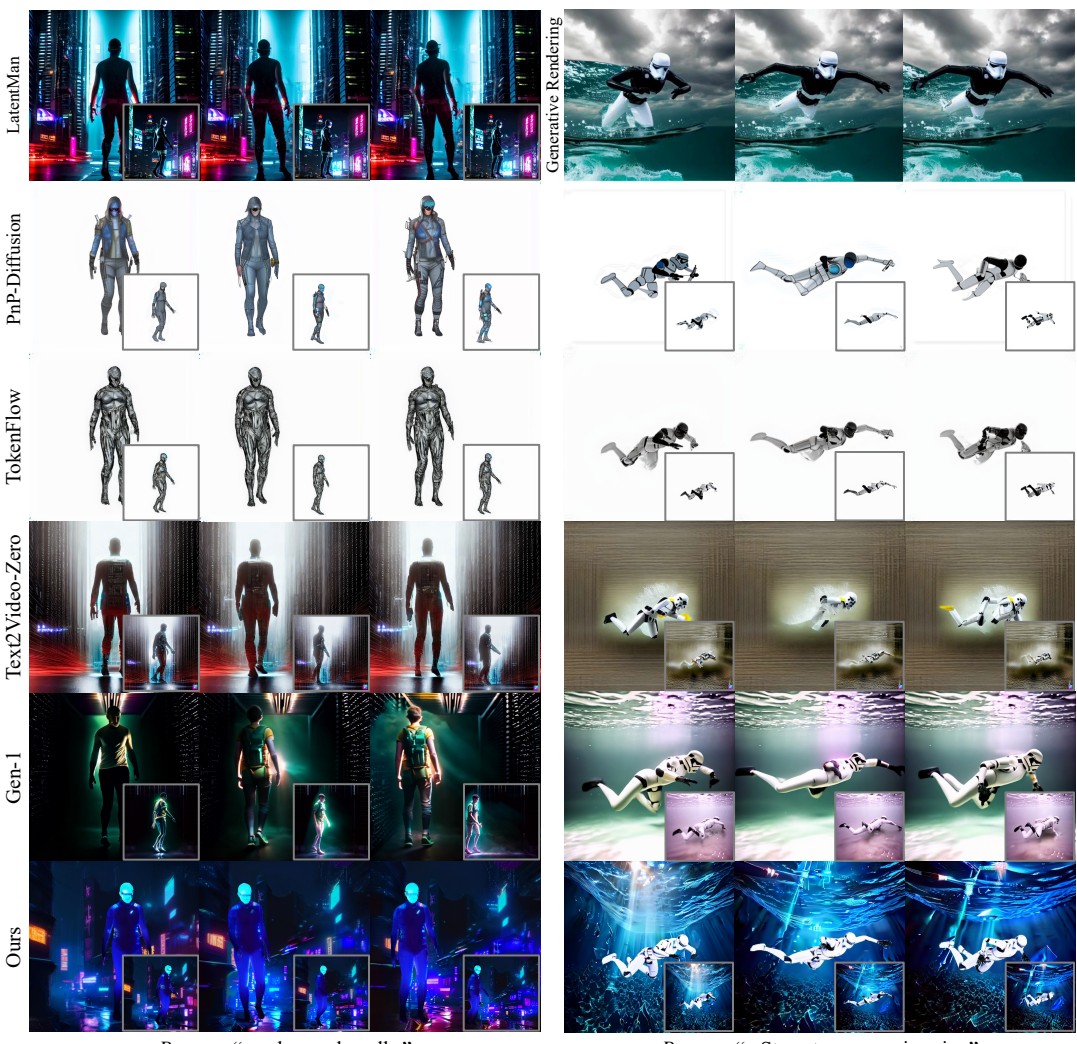

*Prompt:* "a cyberpunk walks"                    *Prompt:* "a Stormtrooper swimming"

Figure 4. **Qualitative comparisons of multi-view video generation.** We compare our method against Text2Video-Zero (Khachatryan et al., 2023), PnP-diffusion (Tumanyan et al., 2023), TokenFlow (Geyer et al., 2023), Gen-1 (Esser et al., 2023), LatentMan (Eldesokey & Wonka, 2024), and Generative Rendering (Cai et al., 2024). We generate videos in the front view and the side view (gray box) on SMPL model data (left column) and Mixamo dataset (right column). Our method manages to generate vivid videos that align with the textual prompts while preserving spatial and temporal consistency.

depth maps) according to their configurations to serve as baselines. These baselines include DDIM inversion-based video stylization methods and video generation methods with different control mechanisms, including PnP-Diffusion (Tumanyan et al., 2023), Text2Video-Zero (Khachatryan et al., 2023), TokenFlow (Geyer et al., 2023), Generative Rendering (Cai et al., 2024), Latent-Man (Eldesokey & Wonka, 2024), and Gen-1 (Esser et al., 2023). PnP-Diffusion is an image style transfer method that is conditioned on the attention feature of the input image by DDIM inversion. We extended the method to stylize videos on a frame-by-frame basis for comparison, aligning with previous work (Geyer et al., 2023). Built upon cross-frame attention, Text2Video-Zero guides the video by warping latents to implicitly enhance video dynamics, and we utilized their official extension, which supports depth control. TokenFlow, Generative Rendering, and LatentMan study frame relations in latent space and establish feature correspondences through nearest neighbor matching, UV maps, and DensePose features, respectively. Gen-1 is a video-to-video model that learns the structure of input videos and transforms the input content (untextured mesh renders) into stylized outputs. Given the availability of the source code for Generative Rendering, we utilize the experimental results presented in their video demos for qualitative comparison.

Table 1. **Quantitative evaluation**. We present FVD values and a comparison highlighting the percentage of user preference for our approach against other methods. Our method shows the best spatio-temporal consistency as measured by the FVD metric (Unterthiner et al., 2018). Users consistently favored Tex4D over all baselines.

| Method | FVD ($\downarrow$) | Appearance Quality | Spatio-temporal Consistency | Consistency with Prompt |
|---|---|---|---|---|
| Text2Video-Zero | 3078.94 | 89.33% | 91.78% | 91.55% |
| PnP-Diffusion | 1390.04 | 86.42% | 87.18% | 89.74% |
| TokenFlow | 1330.43 | 92.31% | 86.84% | 93.42% |
| Gen-1 | 3114.26 | 70.27% | 75.00% | 77.78% |
| LatentMan | 2811.23 | 86.57% | 86.57% | 81.82% |
| Ours | **1303.14** | - | - | - |

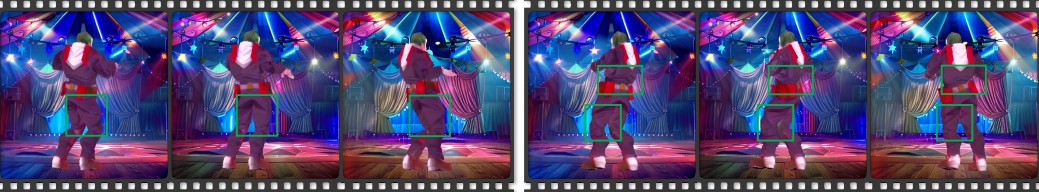

*Prompt: "a Joker dancing in the bright circus stage"*

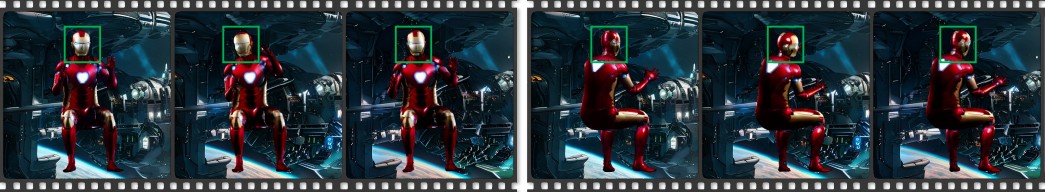

*Prompt: "the Ironman turns steering wheels in the space station"*

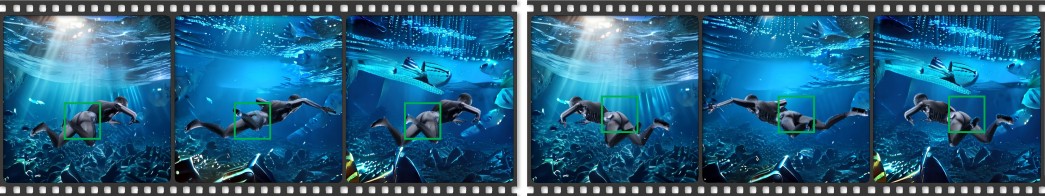

*Prompt: "a machinery swimming in the sea"*

Figure 5. **Qualitative results.** Our method generates multi-view consistent foreground objects with a diverse set of styles and prompts. We highlight the temporal changes in the green boxes.

**Evaluation Metric.** Quantitatively assessing multi-view consistency and temporal coherence is still an unresolved problem. We perform a user study to assess overall performance, including the appearance, temporal coherence and spatial consistency, and the fidelity to prompt based on human preference. In addition, we compute the multi-view coherence via Fréchet Video Distance (FVD) (Unterthiner et al., 2018), a video-level metric that assesses temporal coherence, as utilized in previous approaches (Li et al., 2024; Xie et al., 2024).

## 5.1 QUALITATIVE RESULTS

We present qualitative evaluation in Fig. 4. LatentMan (Eldesokey & Wonka, 2024), Generative Rendering (Cai et al., 2024), TokenFlow (Geyer et al., 2023), and Text2Video-Zero (Khachatryan et al., 2023), which are based on T2I diffusion models with cross-frame attention mechanisms, exhibit significant flickering compared to other methods. This issue arises in part from the empirical and implicit correspondence mapping used to encourage the interframe latent consistency, and the correspondences in the latent space may not exactly match the RGB space. In contrast, our approach interpolates the frames between key frame textures in RGB space, effectively eliminating artifacts caused by latent manipulation. PnP-Diffusion (Tumanyan et al., 2023), which edits frames independently, generates detailed and sophisticated appearances but suffers from poor spatio-temporal consistency due to the loss of temporal correlations in the latent space. While Gen-1 (Esser et al., 2023) (fifth row) produces high-quality videos, it exhibits a jitter effect on the foreground and lacks spatio-temporal consistency.

Furthermore, we present additional multi-view results showcasing a variety of styles and prompts in Fig. 5. Our denoising algorithm, driven by video diffusion models, effectively captures temporal

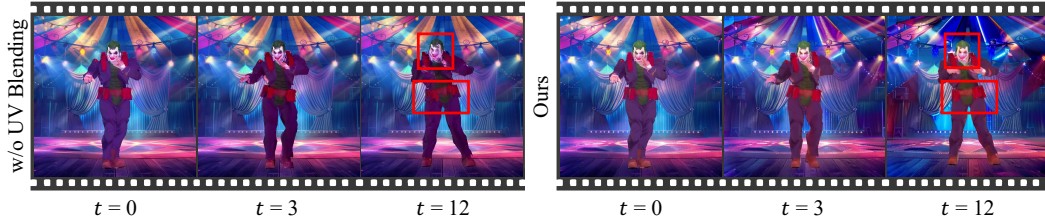

Figure 6. Ablation study on the reference UV blending module. Without this module, the generated textures may lose consistency over time as highlighted in the red boxes.

variations over time. For instance, as highlighted in the green boxes in Fig. 5, our method accurately represents cloth wrinkles (Row 1) and changes in lighting (Rows 2 and 3).

## 5.2 QUANTITATIVE EVALUATION

To quantitatively assess the effectiveness of our proposed method, we follow prior research (Eldesokey & Wonka, 2024; Geyer et al., 2023; Esser et al., 2023) and conduct a comprehensive A/B user study. Our study involved 67 participants who provided a total of 1104 valid responses based on six different scenes drawn from six previous works, with each scene producing videos from two different views. During each evaluation, participants were presented with rendered meshes and depth conditions viewed from two angles, serving as motion references. They were shown a pair of videos: one generated by our approach and the other from a baseline method. Participants were asked to select the method that exhibited superior performance in three criteria: 1) appearance quality, 2) spatial and temporal consistency, and 3) fidelity to the prompts. Table 1 summarizes the preference percentage of our method over the baseline methods. Our method significantly surpasses state-of-the-art methods by a large margin. In addition, our method achieves lower FVD that demonstrates better multi-view consistency in generated video clips.

## 5.3 ABLATION STUDY

**Ablation for texture aggregation.** In Fig. 3 (a) and (b), we present two alternative texture aggregation methods. In Fig. 3 (a), we un-project $\hat{z}_0(z_t)$ and $\epsilon_\theta(z_t)$ into UV space for aggregation. In Fig. 3 (b), we map $z_{t-1}$ to the UV space. Both these two approaches show inferior results compared to our method, which verifies the effectiveness of the proposed texture aggregation algorithm.

**Ablation for UV blending module.** In Sec. 4.4, we propose a reference UV blending schema to resolve the temporal inconsistency caused by latent aggregation. To validate the effectiveness of this mechanism (See Sec. 4.4), we conduct an ablation study by disabling the reference UV blending module (setting $\lambda$ to 0). As shown in Fig. 6, without the UV blending module, our method generates textures with noticeable distortions on the Joker's face over time.

**Ablation for background priors.** Sec. 4.3 discusses the importance of including a plausible background and proposes to learn a dynamic background through diffusion. To verify the effectiveness of this design, we replace the learnable background latents with an all-white background while keeping all other parts unchanged. Fig. 3 (c) illustrates that this ablation experiment produces significantly blurrier textures compared to our full method, highlighting the importance of background learning.

## 6 CONCLUSIONS

In this paper, we present a zero-shot approach that generates multi-view, multi-frame consistent textures for untextured, animated mesh sequences based on a text prompt. By incorporating texture aggregation in the UV space within the diffusion process of a conditional video diffusion model, we ensure both temporal and spatial coherence in the generated textures. To address the variance shift introduced by texture aggregation, we propose a simple yet effective modification to the DDIM sampling algorithm. Additionally, we enhance temporal consistency by introducing a reference UV map and develop a dynamic background learning framework to produce fully textured 4D scenes. Extensive experiments show that our method can synthesize realistic and consistent 4D textures, demonstrating its superiority against state-of-the-art baselines.

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

## A   More Implementation Details

### A.1   Implementation Details

We utilize the CTRL-Adapter (Lin et al., 2024), trained on the video diffusion model I2VGen-XL (Zhang et al., 2023b), as the backbone for generation, with the denoising steps set to $T = 50$. Initially, we center the untextured mesh sequence and pre-define six different viewpoints around the Y-axis in the XZ-plane, uniformly sampled in spherical coordinates, along with an additional top view with an elevation angle of zero and an azimuth angle of $30°$. For latent initialization, we first sample Gaussian noise on the latent textures and then render 2D latent samples for each view to improve the coherence of the generated outputs. During denoising, we upscale the latent resolution to $96 \times 96$ to reduce aliasing. We empirically set the blending coefficient to $0.2$. It takes approximately 30 minutes to generate a video with 24 key frames taken on a RTX A6000 Ada GPU. We decode the denoised latents in key frames to RGB images, and then un-project and aggregate these images to transform the latent UV maps to RGB textures as previous works (Liu et al., 2023b; Cao et al., 2023; Huo et al., 2024). Finally, we interpolate the textures of the key frames at intervals of 3 to synthesize the final video clips.

### A.2   Denoising Algorithm of Our Method

We present the complete workflow of our method in Algorithm 1. The reference UV map $\mathcal{T}_{\mathcal{UV}}$ is constructed by progressively combining latent textures over time, with each new texture filling only the unoccupied texels in the reference UV map.

---

**Algorithm 1** Tex4D

---

**Input:** UV maps $\mathcal{UV} = \{UV_1, ..., UV_K\}$; depth maps $\mathcal{D} = \{D_{1,1}, ..., D_{1,V}, D_{2,1}, ..., D_{K,V}\}$; text prompt $\mathcal{P}$; CTRL-Adapter model $\mathcal{C}$; rendering operation $\mathcal{R}$; unproject operation $\mathcal{R}^{-1}$; cameras $\mathbf{c}$; $T$ diffusion steps; $\mathcal{T}$ latent textures (including foreground and background); $\lambda$ blending weight; $k$ key frames

$\mathcal{T}_T \sim \mathcal{N}(\mathbf{0}, \mathcal{I})$                    // Sample noise in UV space
$\tilde{z}_T, \mathcal{M}_{\text{fg}} = \mathcal{R}(\mathcal{T}_T; \mathbf{c})$
$z_{b,T} \sim \mathcal{N}(\mathbf{0}, \mathcal{I})$
$z = z_T = \tilde{z}_T \odot \mathcal{M}_{\text{fg}} + z_{b,T} \odot (1 - \mathcal{M}_{\text{fg}})$         // Composite latents
**For** $t = T, \dots, 1$ **do**
  $z_{b,t-1} \leftarrow \mathcal{C}(z_{b,t}; \mathcal{D}, \mathcal{P})$
  $\epsilon_\theta \leftarrow \mathcal{C}(z_t; \mathcal{D}, \mathcal{P})$               // Estimate noise from $\mathcal{C}$
  $\hat{z}_0(z_t) = \sqrt{\alpha_t} \cdot z_t - \sqrt{1 - \alpha_t} \cdot \epsilon_\theta$
  $\hat{\mathcal{T}}_0, \mathcal{M}_{\mathcal{UV}} \leftarrow \mathcal{R}^{-1}(\hat{z}_0; \mathbf{c}, \mathcal{UV})$         // Bake textures by Eq. 4
  $\mathcal{T}_{\mathcal{UV}} = \text{Combine}(\hat{\mathcal{T}}_0; \mathcal{M}_{\mathcal{UV}})$
  **For** $k$ in $1, ..., K$ **do**
    $\mathcal{T}_{t-1}^k = \sqrt{\alpha_{t-1}} \cdot \hat{\mathcal{T}}_0^k + \sqrt{1 - \alpha_{t-1}} \left( \sqrt{\frac{\alpha_t}{1-\alpha_t}} \cdot (\sqrt{\alpha_t} \mathcal{T}_t^k - \hat{\mathcal{T}}_0^k) + \sqrt{1 - \alpha_t} \cdot \mathcal{T}_t^k \right)$ // Denoise Eq. 6
    $\mathcal{T}_{t-1}^k = ((1 - \lambda) \cdot \mathcal{T}_{t-1}^k + \lambda \cdot \mathcal{T}_{\mathcal{UV}}) \odot \mathcal{M}_{\mathcal{UV}}^k + \mathcal{T}_{\mathcal{UV}} \odot \left(1 - \mathcal{M}_{\mathcal{UV}}^k\right)$  // Blend textures by Eq. 8
  $\tilde{z}_{t-1}, \mathcal{M}_{\text{fg}} = \mathcal{R}(\mathcal{T}_{t-1}; \mathbf{c}, \mathcal{UV})$
  $z_{t-1} = \tilde{z}_{t-1} \odot \mathcal{M}_{\text{fg}} + z_{b,t-1} \odot (1 - \mathcal{M}_{\text{fg}})$     // Composite latents by Eq. 7
  $z = z_{t-1}$
**Output:** $z$

---

## B   More Qualitative Results

### B.1   Multi-view Results

In Fig. 12, we present additional characters generated by Tex4D, showcasing the method's effectiveness and its ability to generalize across a diverse array of styles and prompts. We also evaluate Tex4D on non-human character animations in Fig. 13, demonstrating its robust generalization capa-

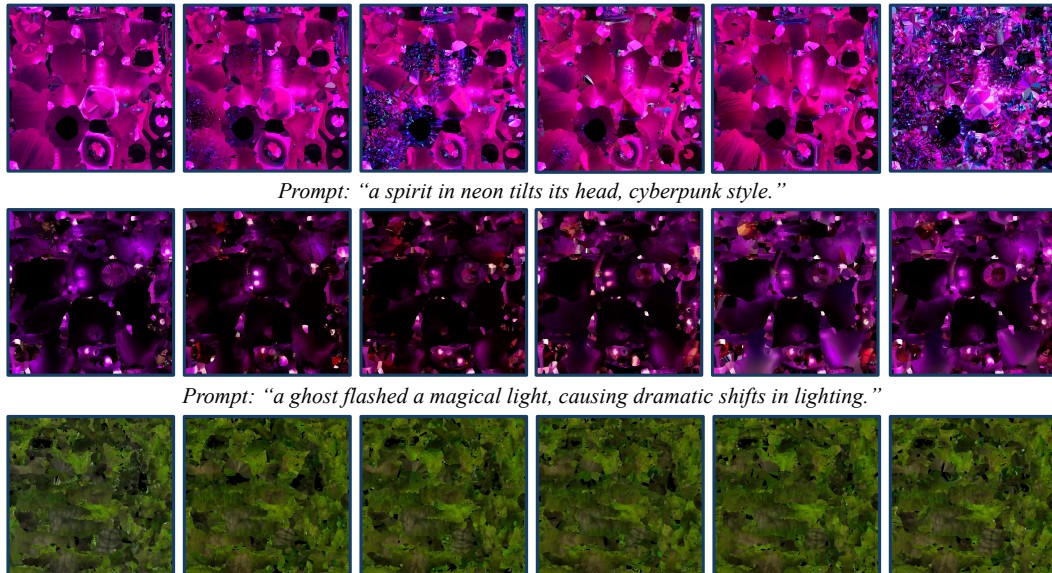

*Prompt: "a spirit in neon tilts its head, cyberpunk style."*

*Prompt: "a ghost flashed a magical light, causing dramatic shifts in lighting."*

*Prompt: "a mossy stone monster dances in a mysterious forest."*

Figure 7. **Visualization of generated textures for mesh sequences.** Our method effectively incorporates temporal changes, such as lighting variations, wrinkles, and appearance transformations, directly into the textures, eliminating the need for post-production by artists.

bilities across various types of mesh sequences. In each case, we provide two different view to show that our method can ensure the multi-view consistency.

To emphasize the temporal changes in the generated textures, we also design some prompts, for example, 'flashed a magical light', 'dramatic shifts in lighting', 'cyberpunk style' in our experiments. As shown in Fig. 13, the results of 'ghost', 'King Boo' and 'Snowman' validate the effectiveness of our method in generating different level of temporal changes by a variety of textual prompts, while maintaining the consistency both spatially and temporally. Additionally, we provide a supplementary video that includes baseline comparisons and multi-view results for all examples.

### B.2 TEXTURE RESULTS

In this section, we present the texture sequences which are the intermediate results of our pipeline. Our method utilizes XATLAS to unwrap the UV maps from meshes without human labors. XATLAS is a widely used library for mesh parameterization and UV unwrapping, commonly integrated into popular tools and engines, facilitating efficient texture mapping in 3D graphics applications. As shown in Fig. 7, our method seamlessly bakes temporal changes, including lighting variations, wrinkles, and appearance transformations, directly into the textures, removing the need for manual post-production by artists.

## C    MORE ABLATION RESULTS

**Ablation on Background**    To show the effects of various background latent initialization strategies, we provide additional examples, including the approach used in the texture synthesis method (Liu et al., 2023b) and a background that contrasts sharply with the foreground object, as shown in Fig. 8. In detail, (Liu et al., 2023a) encodes the backgrounds with alternative random solid color images. For the high contrast background experiment, we use the latents obtained from the DDIM inversion of highly contrast foreground and background to initialize our latents.

**Ablation on Reference UV Blending**    We present an additional ablation study to illustrate how our UV blending module enhances temporal consistency across frames. As shown in Fig. 9, the absence of UV blending results in noticeable distortions, underscoring the importance of this module in maintaining visual coherence.

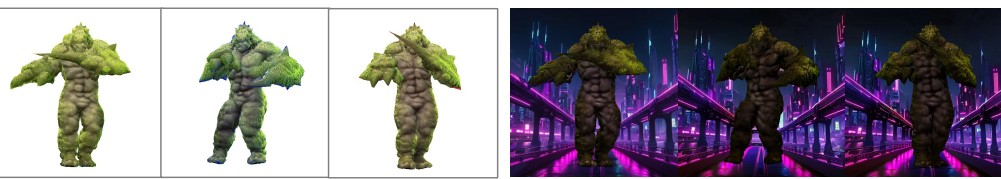

(a) Background w/ alternative random noise        (b) Background highly contrast w/ foreground

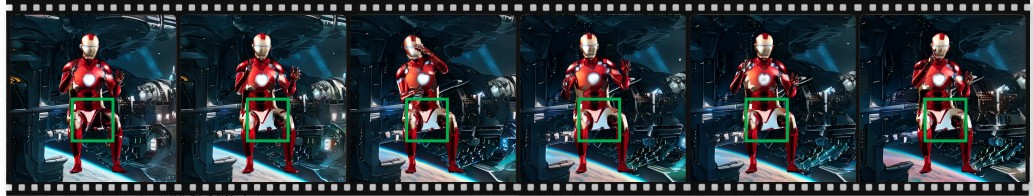

(c) Ours w/o background priors (foreground composited with our background)

Figure 8. **More ablation study on the background priors.**

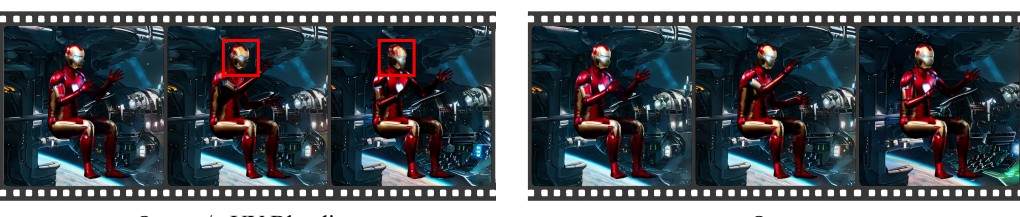

Ours w/o UV Blending                                                    Ours

Figure 9. **More ablation study on the reference UV blending module.**

## D   MORE METHOD COMPARISONS

### D.1   COMPARISON WITH DEPTH-CONDITIONED VIDEO DIFFUSION MODELS

While depth-conditioned video diffusion models are effective at generating visually compelling results from a single viewpoint, they often struggle to maintain consistent multi-view representations of a single object, such as a character, across different perspectives. To highlight this limitation, we present multi-view results from the depth-conditioned video diffusion model in Fig. 11. The primary cause of this issue is that depth conditions are inherently view-dependent, in contrast to UV maps, which provide global information about the 3D space, enabling a unique mapping for each 3D point across all views.

### D.2   COMPARISON WITH TEXTURED MESH ANIMATIONS

In this section, we highlight the differences between our method and traditional approaches, demonstrating the effectiveness of 4D texturing in capturing temporal variations (e.g., lighting and wrinkles) within mesh sequences to produce vivid visual results. Traditional methods typically involve texturing a base mesh (often referred to as a clay mesh) and animating it using skinning techniques. This animated sequence is then refined by technical artists who control scene lighting or simulate cloth dynamics to achieve the final visual presentation. This process is labor-intensive and demands specialized expertise in cinematic production and technical engines.

In contrast, our method offers a streamlined alternative by directly integrating complex temporal changes into mesh sequences. As shown in Fig. 5, 12 and 13, our approach effectively captures intricate temporal effects such as cloth wrinkles, dynamic lighting, and evolving appearances using textual prompts, significantly simplifying the workflow while maintaining high-quality results.

We demonstrate the limitations of traditional textured mesh animation in handling complex temporal changes in Fig. 10. Specifically, we employ the Text2Tex (Chen et al., 2023b) to generate the texture for the input mesh in T-pose and render it from multiple viewpoints. To ensure a fair comparison, we composite the rendered results with the background generated by our method. Notably, the 'ghost'

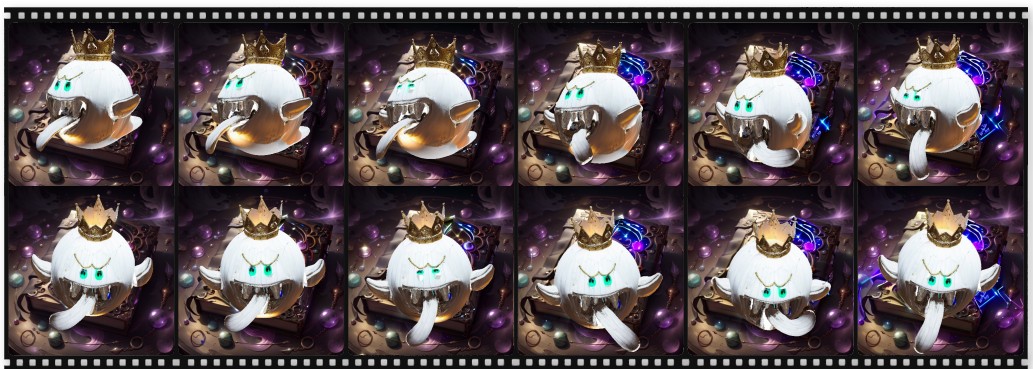

*Prompt: "a ghost flashed a magical light, causing dramatic shifts in lighting."*

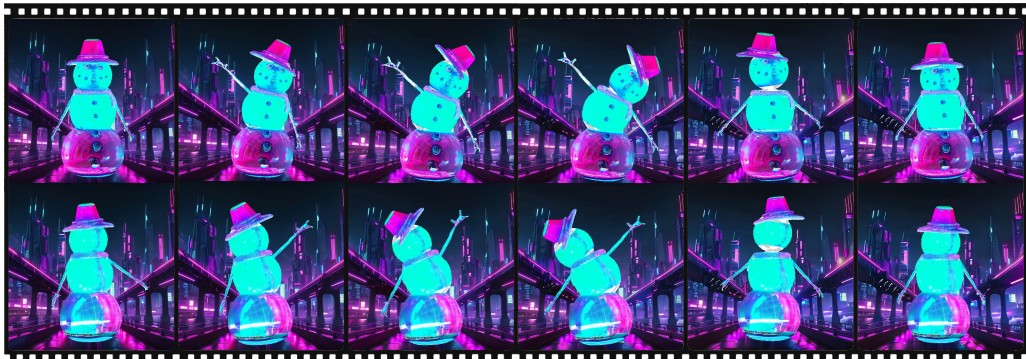

*Prompt: "a spirit in neon tilts its head, cyberpunk style."*

Figure 10. **Results of textured mesh animation (Text2Tex).**

and 'snowman' examples exhibit visible seams during animation due to self-occlusions are common appeared in dynamic poses but cannot be accurately predicted during T-pose texture generation. This results in empty texels and disrupts the visual continuity of the animation.

## E   LIMITATIONS AND DISCUSSION

One limitation of our method is the lack of seamless integration between the generated textures and the background, resulting in a disjointed appearance where the foreground and background elements may seem artificially stitched together. This issue arises due to the absence of a comprehensive scene-level dataset. Alternatively, our approach relies on a shared background mesh across different views, which disrupts overall consistency. Addressing scene-level 4D texturing remains an open challenge that we aim to explore in future work. In addition, we notice that our method is relatively computational intensive compared with other texture synthesis methods. The computation time of our method primarily depends on the foundation model (CTRL-Adapter), which takes approximately 5 minutes to generate a 24-frame video. We anticipate significant efficiency improvements with advancements in conditioned video diffusion models, further enhancing the practicality of our approach.

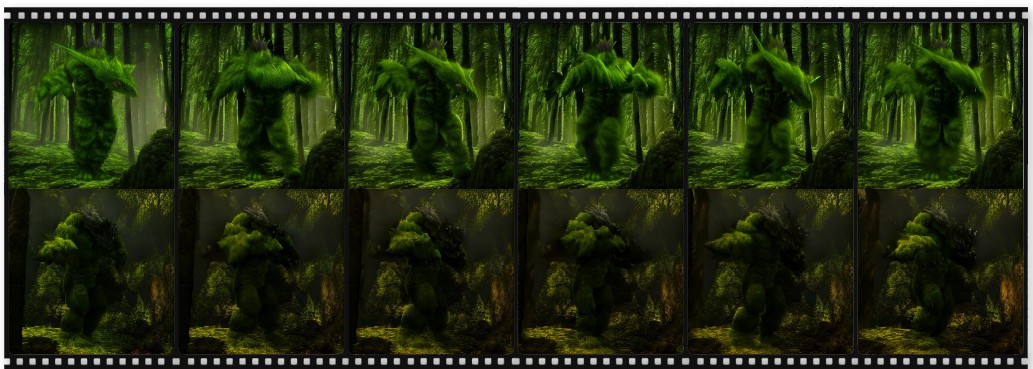

*Prompt: "a mossy stone monster dances in a mysterious forest."*

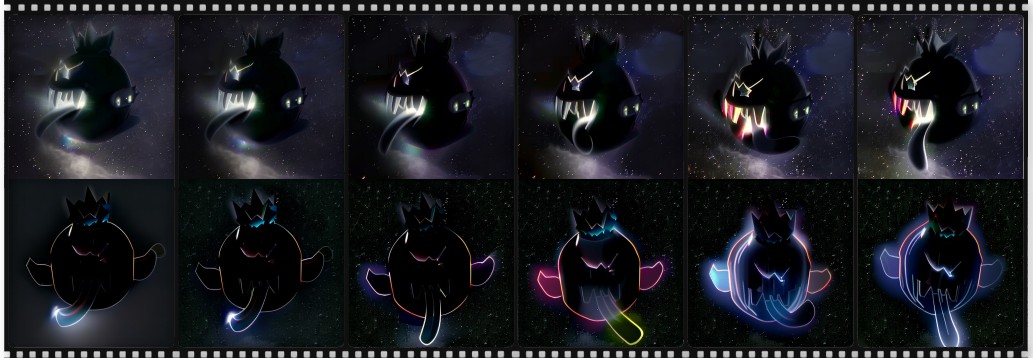

*Prompt: "a ghost flashed a magical light, causing dramatic shifts in lighting."*

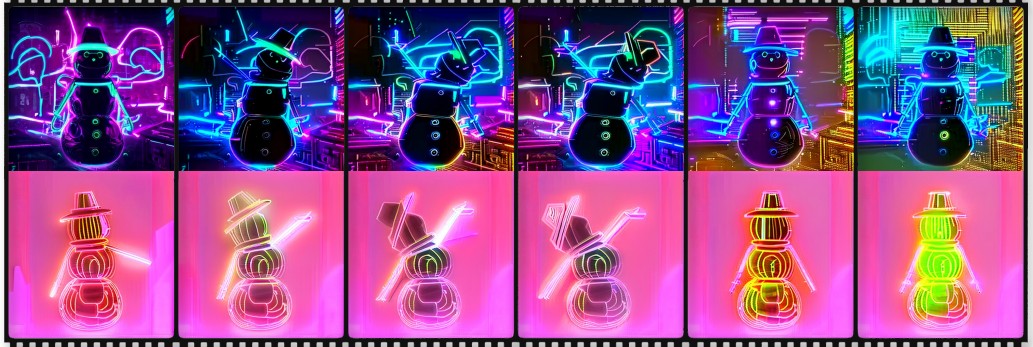

*Prompt: "a spirit in neon tilts its head, cyberpunk style."*

Figure 11. **Multi-view results from video diffusion model.**

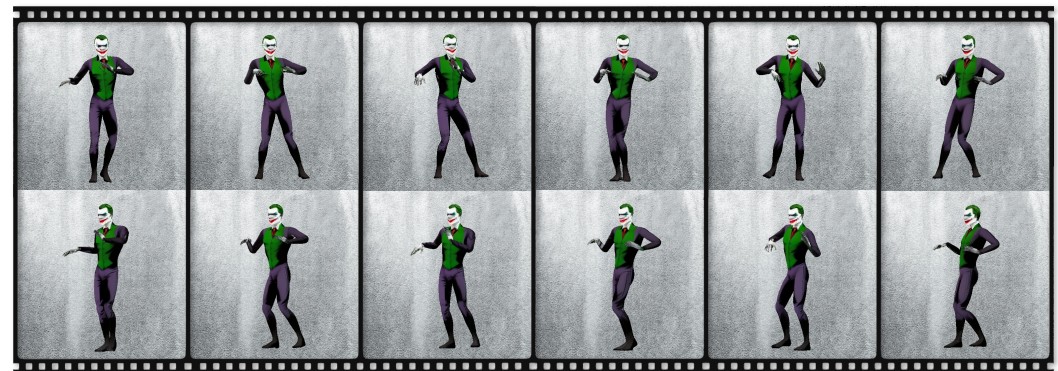

*Prompt: "the Joker dances, comic style"*

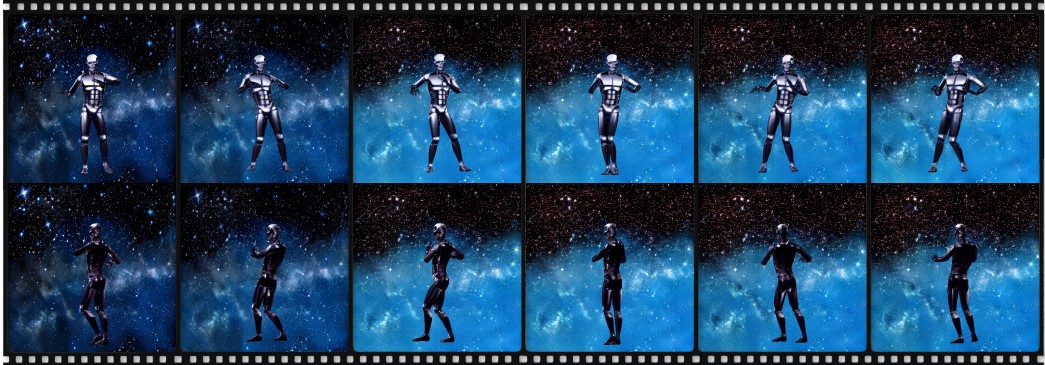

*Prompt: "the terminator dancing in the milky way"*

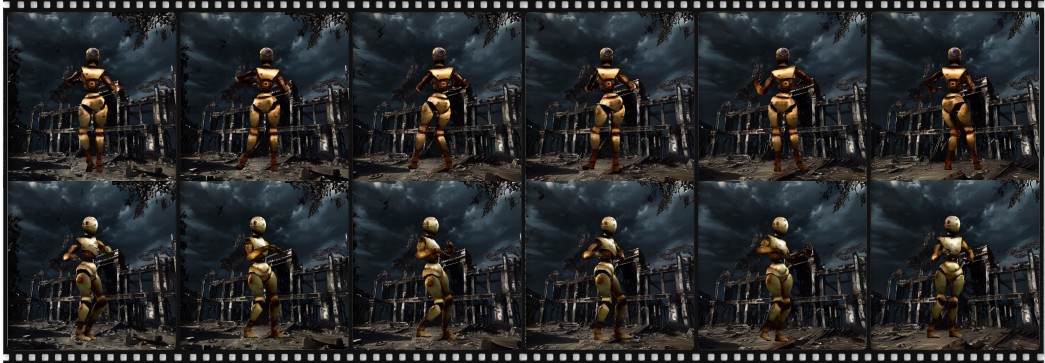

*Prompt: "a rusty robot dances in ruins"*

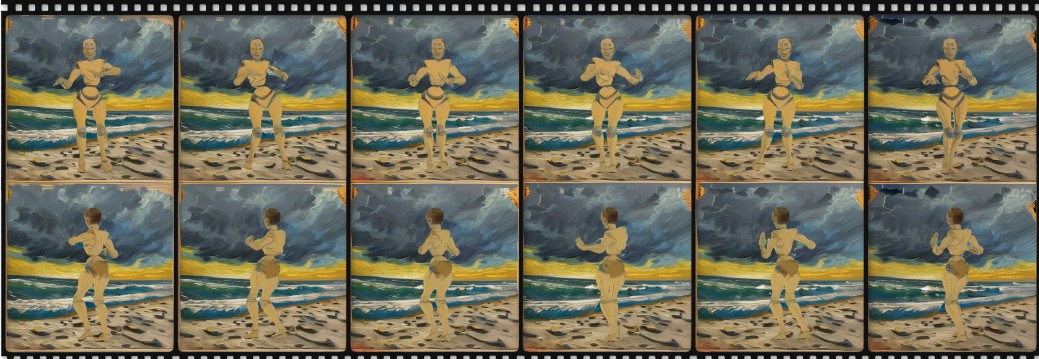

*Prompt: "a sketch of bot dancing in a sandy beach, Van-Gogh style."*

Figure 12. **More qualitative results.**

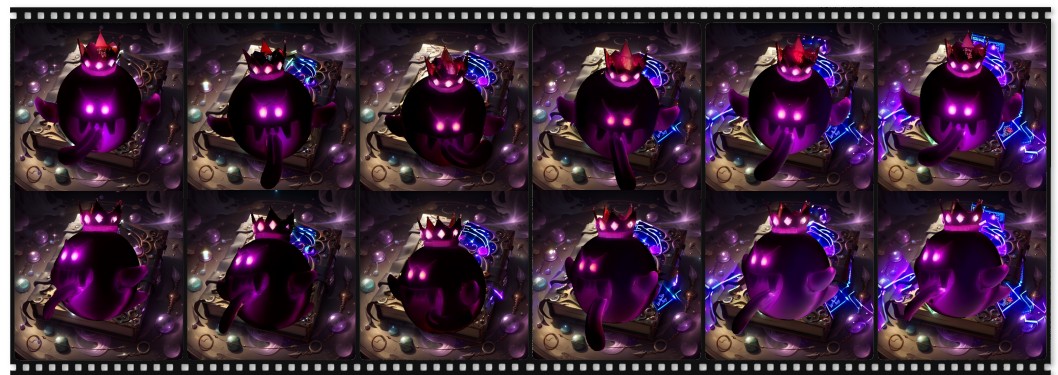

*Prompt: "a ghost flashed a magical light, causing dramatic shifts in lighting."*

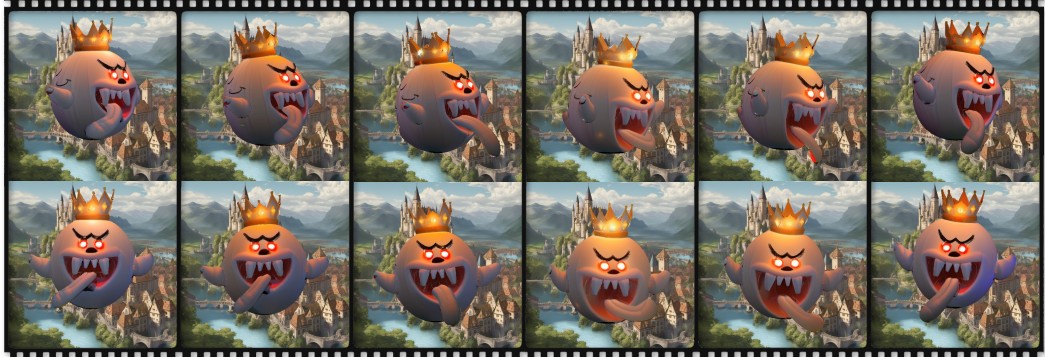

*Prompt: "a dingy, magic King Boo, flashing a weird light, static background."*

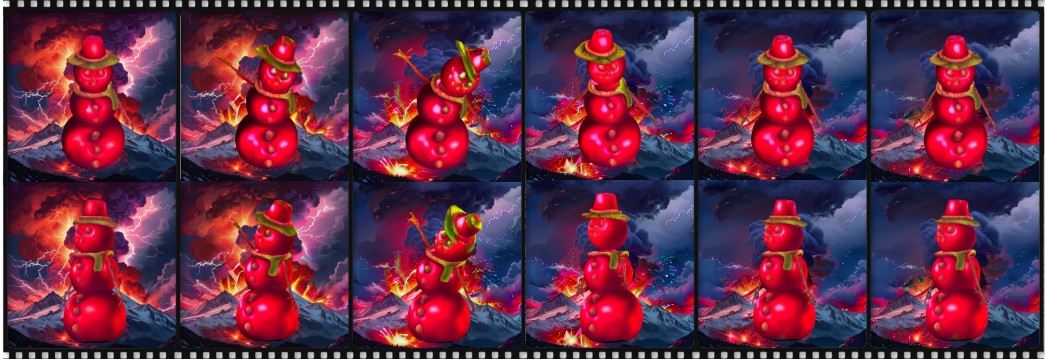

*Prompt: "a sprite of fiery plums tilts its head, in full color."*

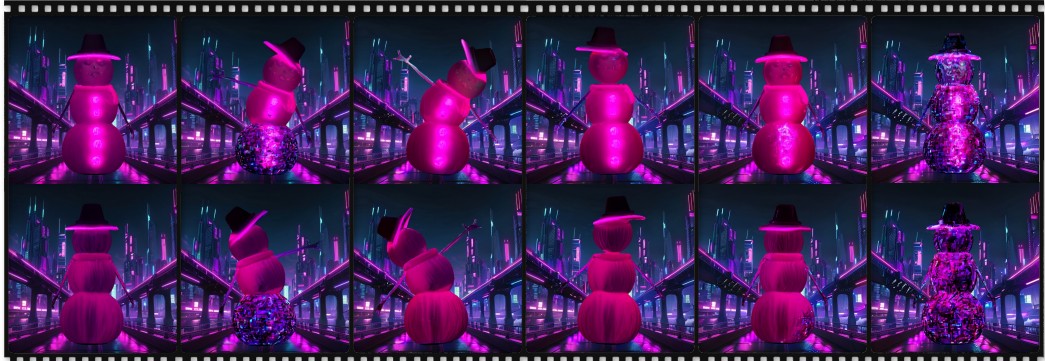

*Prompt: "a spirit in neon tilts its head, cyberpunk style."*

Figure 13. **More qualitative results on non-human character animations.**

