# OpenReview forum: "Tex4D: Zero-shot 4D Scene Texturing with Video Diffusion Models"
_ICLR.cc/2025/Conference — Submitted to ICLR 2025_

### Official Review · Reviewer_CcR5 · 2024-10-24

**Soundness:** 3
**Presentation:** 3
**Contribution:** 2
**Rating:** 5
**Confidence:** 4

**Summary:**

# Summary

This paper focuses on creating temporally consistent and realistic textures for mesh sequences. The input is an untextured mesh sequence and a text prompt. To achieve this, a method named Tex4D is proposed, which is a zero-shot approach that integrates geometry information from mesh sequences with video diffusion models, specifically the depth-conditioned CTRL-Adapter (Lin et al., 2024), to produce multi-view and temporally consistent 4D textures. The model synchronizes the diffusion process across different views through latent aggregation in the UV space. Additionally, a reference latent texture is introduced to strengthen the correlation between frames during the denoising process.

**Strengths:**

# Strengths

- The paper is well-written, making it easy to understand and follow.
- The related works are sufficiently covered.
- Several experiments are conducted for demonstrating its effectiveness.

**Weaknesses:**

Regarding the contribution and motivation, I still wonder about the specific meaning of "4D texturing" for the object and how this differs from a 3D model that is first textured and then animated using skinning techniques. Even for dynamic textures, one could also generate the dynamic texture for a 3D model and then animate the character through skinning. This approach seems useful **if** the mesh is also significantly dynamic, such as with topology changes. Further, when it comes to the video background, I noticed in the supplementary video that there are some dynamic effects from the proposed method, but they are not that significant. Could one first perform 3D texturing and then render it with a generative background video?

I think the following suggestions could be helpful in justifying the significance of the setting in this paper.
- Provide specific examples or use cases where 4D texturing offers advantages over traditional 3D texturing and animation.
- Clarify how the proposed method handles dynamic meshes with topology changes, if applicable.
- Compare the method with a pipeline of 3D texturing followed by rendering with a generative background video, highlighting any benefits of the presented "integrated" approach.

I also have concerns about the novelty of the paper. The entire pipeline can be seen as a depth-conditioned CTRL-Adapter for mesh sequence texturing with UV space aggregation, which feels like a straightforward composition of existing models. I would prefer to see a simple yet effective method of tackling a critical problem. However, as I am still uncertain about the meaning/significance of "4D texture," this makes me somewhat skeptical about the proposed pipeline.

I think the authors could provide some arguments for the novelty of the proposed method:
- Highlight the key technical innovations in the pipeline beyond the composition of existing models.
- Explain how the introduced method addresses specific challenges in 4D texturing that are not easily solved by existing methods.
- Provide a clearer definition and motivation for "4D texturing" to help readers understand its significance in the context of their work (similar to the previous questions).

I understand the difficulty in evaluating the results, but it would be helpful and necessary to conduct an evaluation of Appearance Quality, Spatio-temporal Consistency, and Consistency with Prompt via a user study - The quantitative evaluation is insufficient.

The following action can be taken:
- Conduct a user study evaluating the specific aspects, e.g., Appearance Quality, Spatio-temporal Consistency, and Consistency with Prompt, and compare the proposed method with previous models.

Limitations and Future Works should be included.
- For instance, the authors may discuss the current limitations of their approach, such as some failure cases or more specifically, the types of meshes or textures it struggles with, etc.
- Potential future improvements or extensions to the method.
- Broader implications or applications of this work in related fields.


# Minor Comments

- "To resolve this issue, we analyze the underlying causes and propose a simple yet effective modification to the DDIM (Song et al., 2020) sampling process." In the introduction section, it would be beneficial to briefly explain how you achieved this.

**Questions:**

Q: "While these methods produce multi-view consistent textures for static 3D objects, they do not address the challenge of generating temporally consistent textures for mesh sequences." I would appreciate further clarification on the motivation behind "generating temporally consistent textures" for mesh sequences. Could you provide examples where dynamic 4D texturing is essential and cannot be achieved through traditional methods?

Q: How does the model ensure robustness when dealing with varying UV mapping results?
- How sensitive is the method to different UV unwrapping techniques?
- Did the authors experiment with different UV mapping strategies, and if so, what were the results?
- Are there any limitations or best practices for UV mapping when using this method?

---

> ### Author Response · Authors · 2024-11-21
> **Author Response to Reviewer CcR5 (1/2)**
>
> **Q1:** I still wonder about the specific meaning of "4D texturing" for the object and how this differs from a 3D model that is first textured and then animated using skinning techniques. Even for dynamic textures, one could also generate the dynamic texture for a 3D model and then animate the character through skinning. This approach seems useful if the mesh is also significantly dynamic, such as with topology changes.
>
> **A1:** We thank the reviewer for these insightful comments. "4D texturing" in our work refers to the process of generating temporally and spatially consistent textures for dynamic mesh sequences over time. This differs from traditional pipelines that generate static 3D textures first and subsequently animate the mesh through skinning techniques. While traditional methods can handle static textures effectively, they fall short in scenarios requiring dynamic, time-varying textures that reflect changes such as lighting variations, motion-specific deformations (e.g., wrinkles), or stylized animations. In our additional results, the `snowman` case exhibits significant dynamic changes. We suggest the reviewer kindly refer to our updated paper (**Fig. 13**) and supplementary video.
>
> ---
>
> **Q2:** Could one first perform 3D texturing and then render it with a generative background video? Clarify how the proposed method handles dynamic meshes with topology changes, if applicable. Compare the method with a pipeline of 3D texturing followed by rendering with a generative background video, highlighting any benefits of the presented "integrated" approach. Could you provide examples where dynamic 4D texturing is essential and cannot be achieved through traditional methods?
>
> **A2:** We thank the reviewer for providing this insightful suggestion. Please kindly refer to [General Response (A)](https://openreview.net/forum?id=0Lpz2o6NDE&noteId=wavOGNiB1b) for the discussion with traditional textured mesh animations. We suggest the reviewer go through our updated supplementary video for the dynamic results and comparison with textured mesh animations.
>
> ---
>
> **Q3:** I also have concerns about the novelty of the paper. The entire pipeline can be seen as a depth-conditioned CTRL-Adapter for mesh sequence texturing with UV space aggregation, which feels like a straightforward composition of existing models. Provide a clearer definition and motivation for "4D texturing" to help readers understand its significance in the context of their work.
>
> **A3:** We thank the reviewer for the suggestion to provide a clearer definition and motivation for "4D texturing" and we have added these in our paper (updated in Abstract (**L034-045**) and Introduction (**L057-060, 085-087**)). For the consideration of novelty, our method is the first to handle the task of 4D scene texturing and demonstrate the capability of video diffusion model in offering temporal variations with mesh guidance. In addition, we discuss our motivation in [General Response (B)](https://openreview.net/forum?id=0Lpz2o6NDE&noteId=wavOGNiB1b), and you can kindly refer to it.
>
> ---
>
> **Q4:** I understand the difficulty in evaluating the results, but it would be helpful and necessary to conduct an evaluation of Appearance Quality, Spatio-temporal Consistency, and Consistency with Prompt via a user study - The quantitative evaluation is insufficient.
> > The following action can be taken: Conduct a user study evaluating the specific aspects, e.g., Appearance Quality, Spatio-temporal Consistency, and Consistency with Prompt, and compare the proposed method with previous models.
>
> **A4:** We have already conducted the user study with these metrics in the initial edition (Appearance Quality, Spatio-temporal Consistency, and Consistency with Prompt). Please kindly refer to **Table 1** and **5.2 Quantitative Evaluation (L503-509)** for details.
>
> ---
>
> **Q5:** Limitations and Future Works should be included.
>
> **A5:** Thanks for the suggestion. We provide a "Limitation and Discussion" section in the appendix. We outline the potential limitation in scene-level texturing due to the limited dataset, and leave this for our future work. In addition, we discuss the computation time compared with the 3D texturing method and we believe the computation time may be shorten with the advancement of video diffusion models.

---

> > ### Author Response · Authors · 2024-11-21
> > **Author Response to Reviewer CcR5 (2/2)**
> >
> > **Q6:** How does the model ensure robustness when dealing with varying UV mapping results? How sensitive is the method to different UV unwrapping techniques? Did the authors experiment with different UV mapping strategies, and if so, what were the results? Are there any limitations or best practices for UV mapping when using this method?
> >
> > **A6:** Our method does not rely on a dedicated UV initialization, unlike some texture-painting methods (e.g., Paint3D, Meta TextureGen) edited by artists. Our method utilizes XATLAS to unwrap the UV maps from meshes without human labors. XATLAS is a widely used library for mesh parameterization commonly integrated into popular tools and engines, facilitating efficient texture mapping in 3D graphics applications. We visualize the texture sequences generated by our method in **Fig. 7**.
> >
> > ---
> >
> > **Q7:** Minor Comments: "To resolve this issue, we analyze the underlying causes and propose a simple yet effective modification to the DDIM (Song et al., 2020) sampling process." In the introduction section, it would be beneficial to briefly explain how you achieved this.
> >
> > **A7:** Thanks for the suggestion. We have updated this sentence in our paper (**L092-095**) for a more detailed explanation.
> >
> > ---
> >
> > We sincerely thank the reviewer for providing perceptive comments and suggestions on our work. We have significantly revised our paper based on your valuable suggestions. Specifically, we have added clearer definition and motivation for 4D texturing, and discussed the limitation of traditional textured mesh animations before introducing our method, together with visual comparison with textured mesh animations, texture visualization, and additional discussions. We believe these points will strengthen our work.

---

> ### Author Response · Authors · 2024-11-24
> **Follow-Up on Rebuttal Discussion**
>
> Dear Reviewer CcR5
>
> We sincerely appreciate your reviews and comments on our paper. Since the discussion phase ends on Nov 26, we would like to know whether we have addressed all your concerns. Please let us know if you have further questions after reading our rebuttal.
>
> We hope to address all the potential issues during the discussion period. Thank you once again for your time and effort in reviewing our submission.
>
> Submission#2974 Authors

---

> > ### Comment · Reviewer_CcR5 · 2024-11-27
> > **reply to authors**
> >
> > Thank you for your detailed response. Most of my concerns have been addressed, but I still can not agree with the explanation in A1: "...This differs from traditional pipelines that generate static 3D textures first and subsequently animate the mesh through skinning techniques..." When I referred to "dynamic texture," I meant animatable textures, which could represent true dynamic textures. For clarification, you might refer to some examples available in asset stores. From the video, I feel the texture dynamic patterns are limited (not that dynamic and most results are just with some simple patterns).
> > While I now see some merit in this work as a *"first attempt"* to tackle this problem, I remain unconvinced by the arguments presented in A1. I have adjusted my score from 3 to 5, as some of the answers effectively addressed my concerns. However, overall, I remain critical of the underlying assumptions and approach of this work.

---

> > > ### Author Response · Authors · 2024-11-27
> > >
> > > Thank you for your further clarification and thoughtful suggestions. One of the primary goals of our work is to combine the expressive capabilities of video diffusion models with the consistency inherent in meshes and textures for the creation of consistent and controllable character animations, as the current video diffusion models lack the ability to ensure the multi-view consistency for characters, and the dynamic texture creation is labor-intensive. We also demonstrate that video diffusion models can offer temporal variations for dynamic textures. To the best of our knowledge, no prior work has explored dynamic texture creation using diffusion models in a zero-shot manner.
> > >
> > > We appreciate your feedback and will consider explicitly emphasizing dynamic texture creation (with illustrative examples) as part of our contributions in the revision.

---

### Official Review · Reviewer_jsAN · 2024-11-02

**Soundness:** 2
**Presentation:** 2
**Contribution:** 2
**Rating:** 5
**Confidence:** 4

**Summary:**

This paper introduces 4D scene texturing to generate textures that are consistent both temporally and across multiple views for animated mesh sequences. Tex4D, uses 3D geometry to synchronize diffusion processes, incorporates video generation model insights for temporal consistency, and modifies the DDIM sampling process to improve texture clarity.

**Strengths:**

The paper is the first work to perform video generation based on animated mesh sequences, while its UV mapping strategy ensures multi-view consistency. The experimental results show significant advantages compared to some existing works.

**Weaknesses:**

Under the current pipeline, this work has yielded highly effective results. However, the importance of this pipeline should be further clarified, such as by comparing it with pipelines based on 2D poses or textured meshes. The paper should include more comprehensive comparisons to highlight the contribution of the pipeline. For example, is it a reasonable pipeline to first generate textured meshes and then use animated meshes for video generation?

**Questions:**

Overall, the experimental results are quite satisfactory; however, there is a lack of explanation regarding the advantages of the pipeline compared to other pipelines using 2D poses and textured meshes.

Animation can drive the mesh. Are the position and rotation of the mesh manually controlled, such as in the second example on the first page?

How is the animated mesh obtained? We observe some temporal changes in Figure 5. Is this one of the contributions of your paper? How do you distinguish between temporal changes and temporal inconsistencies, as there are some temporal inconsistencies in your results?

Would using textured meshes yield better outcomes? What are the advantages of generating videos with untextured meshes compared to textured meshes?

What is the difference between using a 2D pose and an animated mesh?

---

> ### Author Response · Authors · 2024-11-21
> **Author Response to Reviewer jsAN**
>
> **Q1:** The importance of this pipeline should be further clarified, such as by comparing it with pipelines based on 2D poses or textured meshes. The paper should include more comprehensive comparisons to highlight the contribution of the pipeline. For example, is it a reasonable pipeline to first generate textured meshes and then use animated meshes for video generation? Would using textured meshes yield better outcomes?
>
> **A1:** We appreciate the reviewer’s insightful comment. To address the concern, we have added visual comparisons of traditional textured mesh animations in **Fig. 10**, accompanied by a detailed video comparison in the supplementary materials. Additionally, our pipeline is designed to handle general characters, not limited to human figures, where 2D poses may not always be available. This flexibility broadens the scope of our approach. To further support this, we have included more visual results in **Fig. 13** and the supplementary videos, showcasing the versatility and contributions of our method in comparison to existing pipelines. We also compared our method with the traditional textured mesh animations (based on Text2Tex), please kindly refer to [General Response (A)](https://openreview.net/forum?id=0Lpz2o6NDE&noteId=wavOGNiB1b).
>
> ---
>
> **Q2:** Animation can drive the mesh. Are the position and rotation of the mesh manually controlled, such as in the second example on the first page? How is the animated mesh obtained?
>
> **A2:** We note that our work assumes animated mesh sequences as inputs and focus on generating plausible dynamic textures, instead of mesh animation. Specifically, we obtain the animations from Mixamo and Sketchfab websites. The Mixamo assets include skeleton rotations, rigging, and skinning weights. We animate the mesh by linear blend skinning based on the pre-defined skeleton hierarchy. The Sketchfab provides complete animations and we extract the vertices and faces using blender software.
>
> ---
>
> **Q3:** We observe some temporal changes in Figure 5. Is this one of the contributions of your paper? What are the advantages of generating videos with untextured meshes compared to textured meshes?
>
> **A3:** Yes, capturing temporal changes is one of the contributions of our paper. Our method is to capture the temporal changes (e.g., lighting controls, appearance transformations) using video diffusion model, which is hard to obtain with textured mesh animations, please kindly refer to [General Response (B)](https://openreview.net/forum?id=0Lpz2o6NDE&noteId=wavOGNiB1b) for our discussion.
>
> ---
>
> **Q4:** How do you distinguish between temporal changes and temporal inconsistencies, as there are some temporal inconsistencies in your results?
>
> **A4:** Temporal changes and temporal inconsistencies are not necessarily in conflict but represent different aspects of temporal dynamics. Temporal changes refer to deliberate and meaningful variations over time, such as lighting shifts, surface deformations (e.g., wrinkles), or stylistic transformations, which are essential for capturing realism and dynamism in animations. Temporal incosistency usually refers to sudden unrealistic changes caused by limitations of the texture generation methods. We suggest the reviewer kindly refer to **Fig. 13** and our supplementary video for these results.

---

> ### Author Response · Authors · 2024-11-24
> **Follow-Up on Rebuttal Discussion**
>
> Dear Reviewer jsAN
>
> We sincerely appreciate your reviews and comments on our paper. Since the discussion phase ends on Nov 26, we would like to know whether we have addressed all your concerns. Please let us know if you have further questions after reading our rebuttal.
>
> We hope to address all the potential issues during the discussion period. Thank you once again for your time and effort in reviewing our submission.
>
> Submission#2974 Authors

---

### Official Review · Reviewer_MURz · 2024-11-03

**Soundness:** 2
**Presentation:** 3
**Contribution:** 2
**Rating:** 5
**Confidence:** 3

**Summary:**

This paper proposed a 4D scene texturing approach by video diffusion models, a zero-shot pipeline for generating temporally and multi-view consistent textures. In order to aggregate multiview latents in UV space, they discovered the issue of "variance shift" caused by their aggregation, and proposed to modify DDIM sampling process to address the issue. By UV blending during denoising steps, the issue of self-occlusion is addressed and synchronized in invisible regions.

**Strengths:**

1. The authors assert that this is the first method developed specifically for 4D scene texturing.
2. The authors introduce a multi-frame consistent texture generation technique, demonstrating improved consistency in results compared to baseline methods.
3. The paper is fluent and well-written, contributing to its readability and overall clarity.

**Weaknesses:**

1. The generated textures do not blend seamlessly with the background, creating a disjointed appearance that resembles separate foreground and background elements stitched together.
2. Despite claims of multi-view consistency, flickering effects are observed across different views, indicating instability in rendering.
3. Some of the compared methods, such as TokenFlow and Text2Video-Zero, do not utilize mesh or depth inputs, making direct comparisons less equitable.

**Questions:**

1. In the paper, the authors mentioned that the mesh texture could be significantly influenced by the background, providing an example with a white background. I’m curious how the generated texture might look if a non-white background was used, especially one that contrasts strongly with the foreground object. How would such a background affect the consistency and quality of the generated texture?

---

> ### Author Response · Authors · 2024-11-21
> **Author Response to Reviewer MURz**
>
> **Q1:** In the paper, the authors mentioned that the mesh texture could be significantly influenced by the background, providing an example with a white background. I’m curious how the generated texture might look if a non-white background was used, especially one that contrasts strongly with the foreground object. How would such a background affect the consistency and quality of the generated texture?
>
> **A1:** We appreciate the insightful question. In **Fig. 8**, we have included additional experiments to demonstrate the effect of varying background conditions. Specifically, we tested an alternative background noise shuffle used in SyncMVD (a) and a highly contrasting background noise initialization (b). The results indicate that the texture tends to be influenced by the high-contrast background latent initialization, leading to deviations from the intended textual prompt. This suggests that while our method can generate textures under various backgrounds, strong contrast can negatively impact both the consistency and alignment of the texture with the desired description.
>
> ---
>
> **Q2:** The generated textures do not blend seamlessly with the background, creating a disjointed appearance that resembles separate foreground and background elements stitched together.
>
> **A2:** We agree with the observation regarding the disjointed appearance between the foreground and background elements. This issue primarily stems from the lack of a comprehensive scene-level dataset for 4D texturing, which constrains our approach. Currently, we use a shared background mesh across different views, which can disrupt overall visual consistency. Addressing the challenge of seamless integration in scene-level 4D texturing is an open problem, and we intend to explore potential solutions in future work to improve coherence between foreground and background elements. We have updated the Limitation and Discussion section in our appendix.
>
> ---
>
> **Q3:** Some of the compared methods, such as TokenFlow and Text2Video-Zero, do not utilize mesh or depth inputs, making direct comparisons less equitable.
>
> **A3:** We acknowledge the concern regarding the fairness of comparisons with methods like TokenFlow and Text2Video-Zero, which do not utilize mesh or depth inputs. Since our work is the first to address 4D scene texturing, there are no existing methods that directly align with our setup. We have endeavored to provide a comprehensive comparison by evaluating current text-to-image (T2I) and text-to-video (T2V) methods under various conditions, **including depth, mesh**, DDIM features, and DensePose features. While depth and mesh conditions are closest to our scenario, other conditions are also relevant and widely discussed in the context of controllable video generation.
>
> ---
>
> In addition, we kindly encourage the reviewer to view our updated supplementary video, which illustrates the advantages of our method in achieving consistent character generation compared to traditional textured mesh animation methods, as discussed in [General Response (B)](https://openreview.net/forum?id=0Lpz2o6NDE&noteId=wavOGNiB1b).

---

> ### Author Response · Authors · 2024-11-24
> **Follow-Up on Rebuttal Discussion**
>
> Dear Reviewer MURz
>
> We sincerely appreciate your reviews and comments on our paper. Since the discussion phase ends on Nov 26, we would like to know whether we have addressed all your concerns. Please let us know if you have further questions after reading our rebuttal.
>
> We hope to address all the potential issues during the discussion period. Thank you once again for your time and effort in reviewing our submission.
>
> Submission#2974 Authors

---

> > ### Comment · Reviewer_MURz · 2024-11-25
> >
> > Dear authors,
> >
> > Thanks for the rebuttals.
> > * For Q2, I would actually doubt the non-seamless blending origins from the direct mask applied in the equation (7). There should be a soft mask applied to foreground-background, rather than just a simple 1s 0s mask.
> > * For Q3, I understand that this is the first work to address 4D scene texturing, however, a straightforward baseline approach could involve using a 4D mesh texturing method to generate the dynamic foreground object, combined with video inpainting to create a consistent background.
> >
> > Given the thoughts above, I would remain my current rating unchanged.

---

> > > ### Author Response · Authors · 2024-11-28
> > >
> > > Dear Reviewer MURz,
> > >
> > > Thanks for your suggestions. We would appreciate it if you could provide more details on why and how the soft mask could be incorporated into our approach. Additionally, we would like to point out that video inpainting methods are typically view-dependent, which limits their ability to provide consistent backgrounds (similar as we discussed in **D.1**).

---

### Official Review · Reviewer_coJ5 · 2024-11-03

**Soundness:** 3
**Presentation:** 2
**Contribution:** 2
**Rating:** 5
**Confidence:** 4

**Summary:**

This paper introduces a novel framework for generating textures for mesh sequences. The authors utilize a depth-conditioned video diffusion model to ensure temporal consistency in videos generated from rendered mesh sequences for each predefined viewpoint. To achieve multi-view consistency, they adopt a UV space texture aggregation strategy. Additionally, they propose a modified sampling approach to address the issue of blurriness in the generated textures.

**Strengths:**

See below.

**Weaknesses:**

See below.

**Questions:**

Although the experiments provide some evidence of the proposed method’s effectiveness, several concerns remain:

1. Could the authors provide additional qualitative and quantitative comparisons in the ablation study? With only one demonstration, it is difficult to convincingly assess the effectiveness of the proposed method.

2. The authors suggest that video diffusion models struggle with multi-view consistent texturing for 3D mesh sequences due to a lack of 3D geometry awareness. However, the approach already uses a depth-aware video diffusion model, which inherently includes some geometric awareness. Why does this straightforward combination not achieve the desired consistency? Does this imply that depth-aware video diffusion models alone cannot guarantee multi-view consistency even with depth information? If so, could the authors provide performance metrics or visual comparisons showing results when using only the depth-conditioned video diffusion model as a prior? Additionally, for a single viewpoint, does the video diffusion model produce temporally consistent results? If not, visual examples would help clarify.

3. Since a mesh input is available, a straightforward approach could be to texture the mesh on the first frame using methods like Text2Tex or SceneTex, then animate the textured mesh. This method might improve efficiency and naturally maintain multi-view consistency across frames. How does this alternative approach compare in terms of both methodology and performance? An in-depth discussion of these differences would be beneficial.

4. The authors mention that for each predefined viewpoint, a sequence of K rendered meshes is used as input and individually textured by the depth-guided diffusion model. Could the authors clarify the motivation behind this setup? Since the videos are generated separately for each view, multi-view inconsistencies are expected. Why introduce this setup if it inherently leads to consistency issues at the start?

5. While using UV textures for each mesh can enhance multi-view consistency, this approach seems more like an averaging of multiple viewpoints to produce a smoother result. Can the authors elaborate on how this averaging mechanism ensures true multi-view consistency?

6. Given that the current method requires rendering V views for each mesh in the sequence, which may be computationally intensive, could the authors discuss the efficiency of the method? Details on the time required to process a sample would help assess the method's practicality.

7. It would be beneficial to include video visualizations or comparative examples to further illustrate the method's performance and effectiveness.

---

> ### Author Response · Authors · 2024-11-21
> **Author Response to Reviewer coJ5 (1/2)**
>
> **Q1:** Additional qualitative and quantitative comparisons in the ablation study. With only one demonstration, it is difficult to convincingly assess the effectiveness of the proposed method.
>
> **A1:** We follow Reviewer `coJ5` and `MURz`'s suggestion and include one more example for UV reference module ablation in **Fig. 9** and more ablation experiments on backgrounds in **Fig. 8** (alternative noise shuffle strategy used in SyncMVD (a), highly contrast with foreground (b), another case with `Ironman` \(c\)). Our findings suggest that the background initialization indeed affects the appearance of the texture and the background shuffling strategy used in static texture generation may not be applicable in our pipeline.
>
> ---
>
> **Q2:** The authors suggest that video diffusion models struggle with multi-view consistent texturing for 3D mesh sequences due to a lack of 3D geometry awareness. However, the approach already uses a depth-aware video diffusion model, which inherently includes some geometric awareness.
> Why does this straightforward combination not achieve the desired consistency? Does this imply that depth-aware video diffusion models alone cannot guarantee multi-view consistency even with depth information? If so, could the authors provide performance metrics or visual comparisons showing results when using only the depth-conditioned video diffusion model as a prior? Additionally, for a single viewpoint, does the video diffusion model produce temporally consistent results? If not, visual examples would help clarify.
>
> **A2:** We appreciate the reviewer’s observation regarding the role of depth-aware video diffusion models in achieving multi-view consistency. While depth maps indeed provide some geometric information and allow consistent results within a single viewpoint (as demonstrated by the CTRL-Adapter), they are inherently view-dependent and lack global spatial information encoded by the 3D mesh, such as the absolute position of points in a world coordinate system. This limitation prevents consistent character generation and texture generation across different views. In **Fig. 11**, we have added three experiments to demonstrate that depth-conditioned diffusion models alone fail to ensure a global consistent texture. The results highlight how depth maps, despite their localized geometric guidance, cannot resolve inconsistencies across different viewpoints.
> In contrast, UV maps act as a global constraint by mapping 3D spatial positions to consistent 2D UV coordinates $(x, y, z)\_{xyz} \rightarrow (x’, y’)\_{\text{UV}}$, leveraging the point correspondence of the same points on the object in different views, thus encouraging 3D consistency.
>
> ---
>
> **Q3:** Since a mesh input is available, a straightforward approach could be to texture the mesh on the first frame using methods like Text2Tex or SceneTex, then animate the textured mesh. This method might improve efficiency and naturally maintain multi-view consistency across frames. How does this alternative approach compare in terms of both methodology and performance? An in-depth discussion of these differences would be beneficial.
>
> **A3:** We appreciate the reviewer's suggestion for in-depth discussion of textured mesh animation, using methods such as Text2Tex for comparison. Please kindly refer to [General Response (A)](https://openreview.net/forum?id=0Lpz2o6NDE&noteId=wavOGNiB1b) for the discussion with traditional textured mesh animations (Text2Tex) and [General Response (B)](https://openreview.net/forum?id=0Lpz2o6NDE&noteId=wavOGNiB1b) for the motivation of 4D scene texturing.
>
> ---
>
> **Q4:** The authors mention that for each predefined viewpoint, a sequence of K rendered meshes is used as input and individually textured by the depth-guided diffusion model. Could the authors clarify the motivation behind this setup? Since the videos are generated separately for each view, multi-view inconsistencies are expected. Why introduce this setup if it inherently leads to consistency issues at the start?
>
> **A4:** We aim to texuring 4D scenes and capturing temporal variations (e.g., lighting and wrinkles) within mesh sequences to produce vivid visual result, which is widely expected in downstream tasks (e.g., generating a consistent character). Depth maps alone, even when used with video diffusion models, fail to achieve this goal due to their lack of global consistency as shown in **Fig. 13**. Instead, we use UV space as a global reference which is an off-the-shelf attribute of mesh and ensures consistent texturing across all views. Our method merges the appearance from different views and aggregate the information in UV space, and render the latents from the latent UV for each view to achieve the consistency. This strategy provides a robust solution for multi-view consistency while capturing dynamic temporal details, although the initial views may not be well-aligned across different views.

---

> ### Author Response · Authors · 2024-11-21
> **Author Response to Reviewer coJ5 (2/2)**
>
> **Q5:** While using UV textures for each mesh can enhance multi-view consistency, this approach seems more like an averaging of multiple viewpoints to produce a smoother result. Can the authors elaborate on how this averaging mechanism ensures true multi-view consistency?
>
> **A5:** The UV space serves as a global reference and we hope to merge the information from different views to the UV space, which ensures the global consistency. Inspired by some mesh texturing methods (e.g., SyncMVD, Meta TexureGen), we simply uses the weighted aggregation to merge latents observed from different views.
>
> ---
>
> **Q6:** Given that the current method requires rendering V views for each mesh in the sequence, which may be computationally intensive, could the authors discuss the efficiency of the method? Details on the time required to process a sample would help assess the method's practicality.
>
> **A6:** We have included average computation times in the appendix for clarity. Our method requires approximately **30 minutes** per sequence, which is comparable to static texture generation methods like Text2Tex (**22 minutes** for a static texture). The computation time primarily depends on the foundation model (CTRL-Adapter), which takes approximately **5 minutes** to generate a video with 24 frames. We anticipate significant efficiency improvements with advancements in conditioned video diffusion models, further enhancing the practicality of our approach.
>
> ---
>
> **Q7:** It would be beneficial to include video visualizations or comparative examples to further illustrate the method's performance and effectiveness.
>
> **A7:** We have provided additional experiments highlighting temporal changes using textual prompts such as `flashed a magical light`, `dramatic shifts in lighting`, `cyberpunk style` in **Fig. 13** and included these results along with video visualizations in the supplementary materials. Also, we have updated the main paper with **Fig. 7~Fig. 13** for comprehensive comparisons.

---

> ### Author Response · Authors · 2024-11-24
> **Follow-Up on Rebuttal Discussion**
>
> Dear Reviewer coJ5
>
> We sincerely appreciate your reviews and comments on our paper. Since the discussion phase ends on Nov 26, we would like to know whether we have addressed all your concerns. Please let us know if you have further questions after reading our rebuttal.
>
> We hope to address all the potential issues during the discussion period. Thank you once again for your time and effort in reviewing our submission.
>
> Submission#2974 Authors

---

> > ### Comment · Reviewer_coJ5 · 2024-12-01
> >
> > I appreciate the authors' efforts, but my concerns remain unresolved:
> >
> > 1.	The authors provide a few examples to demonstrate the method's effectiveness, but these are not convincing without quantitative experiments on a larger test set, as done in Table 1.
> > 2.	In Fig. 11, the authors present three examples of videos generated by the video diffusion model from different viewpoints. While I agree that there is significant inconsistency between viewpoints, I question whether using such highly inconsistent videos as priors is a reasonable approach. This raises further concerns mentioned in Q5: using UV textures for each mesh to enhance multi-view consistency seems to average out the high inconsistency, resulting in smoother and less detailed textures.
> > 3.	The authors show two examples generated by Text2Tex in Fig. 10. I acknowledge that these examples contain artifacts due to self-occlusions and other factors. How do the results from the proposed method compare for the same examples? Will it provide more detail and texture variation? A direct comparison would offer more insight.
> > 4.	As noted in A2, the authors claim that the current setup is intended to achieve multi-view consistent results. However, given the large variations (as shown in Fig. 11) on different view points by the diffusion model, it’s hard to believe that the proposed method can produced high-quality results. The produced consistent results are just an average of different view points. Of course, the averaged results will be consistent but at the cost of losing details in original individual view points and over-smoothness.
> > 5.	Please refer to A2 and A4 for further clarification.
> >
> > The revisions have not fully addressed my concerns, and the visualizations in Fig. 11 furthter increased my concerns about the plausibility of the paper’s setup (See A2, A4). Given these ongoing concerns, I will maintain my current rating.

---

> ### Author Response · Authors · 2024-11-30
> **Please urge reviewers to participate in discussion**
>
> Dear AC and SAC,
>
> We have provided answers and explanations to reviewers' questions. As some reviewers have not engaged in discussions despite several follow-up emails, can you send an email to urge them to participate before the deadline on Dec 2?
>
> Thank you,

---

> ### Author Response · Authors · 2024-12-01
>
> Thank you for your suggestions and detailed feedback. We would like to further clarify the points you raised.
>
> 1. We appreciate your suggestion regarding additional examples. We will provide more visual examples and include quantitative results from our user study in the appendix in our revision.
>
> 2. (and 4) We think the main concern from the reviewer's side is about the plausibility of the latent aggregation strategy. There are some prior works have demonstrated the effectiveness of generating high-quality textures for static 3D meshes in the T-pose using the texture aggregation in the latent space, such as SyncMVD [1] and Meta 3D TextureGen [2]. Our method denoises the latents in temporal batches and integrates with the multi-view aggregation strategy to maintain the consitency. In addition, we have experimented the direct latents blending strategy in Fig. 3, which causes the oversmooth problem because of the "variation shifts problem" discussed in L305-309. We overcome this issue by rewriting the denoising formula. As shown in the qualitative results, our method produces detailed texture sequences (Fig. 7) and consistent visual appearances (Fig. 12 and Fig. 13).
>
> 3. Thanks for the suggestion. We have included a video comparison in our supplementary material and since these examples are generated using the same assets, they already appear in Fig. 1 (1st case) and Fig. 13 (2nd and 3rd cases), respectively. As such, we refrained from duplicating these comparisons in the main text.
>
> Hope these clarifications address your concerns. Thank you again for your detailed review and for engaging with our work.
>
> ---
>
> **References**
>
> *[1] Yuxin Liu, Minshan Xie, Hanyuan Liu, Tien-Tsin Wong. Text-Guided Texturing by Synchronized Multi-View Diffusion. SIGGRAPH Asia. 2024.*
>
> *[2] Raphael Bensadoun and Yanir Kleiman and Idan Azuri and Omri Harosh and Andrea Vedaldi and Natalia Neverova and Oran Gafni. Meta 3D TextureGen: Fast and Consistent Texture Generation for 3D Objects. arXiv preprint arXiv:2407.02430*

---

### Author Response · Authors · 2024-11-21
**General Response**

### (A) Comparison with Textured Mesh Animations (Reviewer `coJ5`, `jsAN`, `CcR5`)

We follow the insightful suggestions proposed by reviewers and conduct additional experiments to compare our method with the Text2Tex on animated meshes. Text2Tex is a static texture synthesis method. We first use Text2Tex to generate the textures in T-pose and then animate it. As shown in **Fig. 10** and our supplementary video, Text2Tex struggles to produce plausible temporal variations. Furthermore, the `ghost` and `snowman` examples generated by Text2Tex exhibit visible seams. This is because the texture synthesized in T-pose may not cover the entire object due to self-occlusions. As a result, the object will present seam artifacts when animated. Instead, our method can generate vivid consistent characters by textual prompts as shown in **Fig. 13**.

---

### (B) Motivation for 4D Scene Texturing (Reviewer `coJ5`, `jsAN`, `CcR5`)

Our objective is to texture 4D scenes while capturing temporal variations, such as lighting changes, wrinkles, dynamic effects to produce vivid visual results—a key requirement in downstream tasks like character generation.
We agree that texturing a mesh and subsequently animating it is a straightforward approach that aligns with traditional graphics pipelines. However, this approach involves significant post-processing steps, such as lighting adjustments and appearance transformations, to achieve the final visual quality. These steps are labor-intensive and require specialized expertise by artists. Our goal is to alleviate these challenges using video diffusion models.
To emphasize the temporal changes in the generated textures, we also have designed some prompts, for example, `flashed a magical light`, `dramatic shifts in lighting`, `cyberpunk style` in our experiments (**Fig. 13** and **updated video**). Manually creating these sequences would require per-frame texture design, which is much less efficient compared to our method.

---

**References**


*[1] Chen, Dave Zhenyu and Siddiqui, Yawar and Lee, Hsin-Ying and Tulyakov, Sergey and Nießner, Matthias. Text2tex: Text-driven texture synthesis via diffusion models. Proceedings of the IEEE/CVF International Conference on Computer Vision (ICCV). 2023.*

---

### Meta-Review · Area_Chair_indD · 2024-12-18

**Metareview:**

This paper introduces a method for generating temporal and multi-view consistent textures for a mesh sequence in a training-free manner using a pretrained depth-conditioned video diffusion model. The proposed method builds upon previous 3D texture generation methods such as SyncMVD and TexGen, but adapts these ideas for 4D texture generation. The authors propose generating the foreground and background separately and then aggregating them. They also suggest a method to improve the denoising process and reduce blurry results. The experimental results include comparisons with previous methods, demonstrating a lower FVD score of the proposed method.

All reviewers gave negative feedback and were not convinced about the need for a 4D generation method, as it is also possible to generate 3D textures and animate the object. While the authors emphasized in the rebuttal that 4D generation can incorporate lighting changes, wrinkles, and dynamic effects, the additional results were not convincing to all the reviewers and the AC. Furthermore, although the authors claim that applying these changes in the graphics pipeline requires significant labor, if users can animate a mesh, changing lighting and adding dynamic effects would not be much more burdensome.

Regarding the technical contributions, from the AC's perspective, the proposed method appears to be a straightforward extension of the 3D texture generation approach to 4D using a video generative model, with some engineering techniques such as foreground/background separation. However, it does not seem to provide a sufficient technical contribution to meet the ICLR standards.

**Additional Comments On Reviewer Discussion:**

Please see the Metareview.

---

### Decision · Program_Chairs · 2025-01-22

Reject